



# Airborne limb-imaging measurements of temperature, HNO₃, O₃, ClONO₂, H₂O and CFC-12 during the Arctic winter 2015/16: characterization, in-situ validation and comparison to Aura/MLS

Sören Johansson[1], Wolfgang Woiwode[1], Michael Höpfner[1], Felix Friedl-Vallon[1], Anne Kleinert[1], Erik Kretschmer[1], Thomas Latzko[1], Johannes Orphal[1], Peter Preusse[2], Jörn Ungermann[2], Michelle L. Santee[3], Tina Jurkat-Witschas[4], Andreas Marsing[4], Christiane Voigt[4], Andreas Giez[5], Martina Krämer[2], Christian Rolf[2], Andreas Zahn[1], Andreas Engel[6], Björn-Martin Sinnhuber[1], and Hermann Oelhaf[1]

[1]Institute of Meteorology and Climate Research, Karlsruhe Institute of Technology, Karlsruhe, Germany
[2]Institute of Energy and Climate Research - Stratosphere (IEK-7), Forschungszentrum Jülich, Jülich, Germany
[3]Jet Propulsion Laboratory, California Institute of Technology, Pasadena, California, USA
[4]Institute of Atmospheric Physics, Deutsches Zentrum für Luft- und Raumfahrt, Oberpfaffenhofen, Germany
[5]Flight Experiments, Deutsches Zentrum für Luft- und Raumfahrt, Oberpfaffenhofen, Germany
[6]Institute for Atmospheric and Environmental Sciences, Goethe-University Frankfurt, Frankfurt, Germany

*Correspondence to:* S. Johansson (soeren.johansson@kit.edu)

**Abstract.** The Gimballed Limb Observer for Radiance Imaging of the Atmosphere (GLORIA) was operated on board the German High Altitude and LOng range (HALO) research aircraft during the PGS (POLSTRACC/GW-LCYCLE/SALSA) aircraft campaigns in the Arctic winter 2015/2016. Research flights were conducted from 17 December 2015 until 18 March 2016 between 80°W - 30°E longitude and 25°N - 87°N latitude. From the GLORIA infrared limb emission measurements, two dimensional cross sections of temperature, HNO₃, O₃, ClONO₂, H₂O and CFC-12 are retrieved. During 15 scientific flights of the PGS campaigns the GLORIA instrument measured more than 15000 atmospheric profiles at high spectral resolution. Dependent on flight altitude and tropospheric cloud cover, the profiles retrieved from the measurements typically range between 5 and 14 km, and vertical resolutions between 400 m and 1000 m are achieved. The estimated total (random and systematic) 1-$\sigma$ errors are in the range of 1 to 2 K for temperature and 10 % to 20 % relative error for the discussed trace gases. Comparisons to in-situ instruments deployed on board HALO have been performed. Over all flights of this campaign the median differences and median absolute deviations between in-situ and GLORIA observations are -0.75 K $\pm$ 0.88 K for temperature, -0.03 ppbv $\pm$ 0.85 ppbv for HNO₃, -3.5 ppbv $\pm$ 116.8 ppbv for O₃, -15.4 pptv $\pm$ 102.8 pptv for ClONO₂, -0.13 ppmv $\pm$ 0.63 ppmv for H₂O and -19.8 pptv $\pm$ 46.9 pptv for CFC-12. These differences are mainly within the expected performances of the cross-compared instruments. Events with stronger deviations are explained by atmospheric variability and different sampling characteristics of the instruments. Additionally comparisons of GLORIA HNO₃ and O₃ with measurements of the Aura Microwave Limb Sounder (MLS) instrument show highly consistent structures in trace gas distributions and illustrate the potential of the high spectral resolution limb-imaging GLORIA observations for resolving narrow mesoscale structures in the UTLS.



# 1 Introduction

The region of the upper troposphere and lower stratosphere (UTLS) is a key region for climate on Earth (Gettelman et al., 2011). The extra-tropical UTLS is influenced by vertical downward transport from the stratosphere by the Brewer-Dobson circulation, by horizontal transport from the upper tropical troposphere by isentropic mixing, by convective overshooting and

mixing with the troposphere (Holton et al., 1995; Bönisch et al., 2011). The UTLS is challenging to observe. Isentropic mixing in this region happens via very long filaments with small vertical and horizontal extent (Konopka and Pan, 2012). This requires a large horizontal coverage on the one-hand and high spatial resolution on the other hand. Aircraft infrared limb-emission measurements can fill the gap between airborne in-situ instruments and space-borne remote sensing satellites. Airborne in-situ instruments provide a high accuracy, high temporal resolution and along-track sampling, but they are limited to the vertical and

horizontal dimensions of the aircraft's flight track. Space-borne measurements provide global coverage but are limited in terms of spatial sampling and accuracy. Aircraft and balloon measurement campaigns with infrared limb-emission remote sensing instruments have been a source of vertically, spatially and/or temporally resolved observations of temperature and a wealth of trace gases (e.g. Piesch et al., 1996; Friedl-Vallon et al., 2004) as well as important steps for demonstration of technology for future satellite missions (e.g. Fischer et al., 2008).

The Gimballed Limb Observer for Radiance Imaging of the Atmosphere (GLORIA) instrument (Friedl-Vallon et al., 2014) continues the heritage of the series of MIPAS (Michelson Interferometer for Passive Atmospheric Sounding (Fischer and Oelhaf, 1996; Piesch et al., 1996; Friedl-Vallon et al., 2004)) and CRISTA (CRyogenic Infrared Spectrometers and Telescopes for the Atmosphere (Offermann et al., 1999; Ungermann et al., 2012)) instruments. One major improvement of GLORIA compared to its limb-scanning precursors is the usage of an imaging array detector for significantly higher spatial sampling

and precise relative pointing. A wider overview of scientific objectives for these altitudes and the potential of the GLORIA instrument for such science questions is given by Riese et al. (2014).

Measurements with the airborne MIPAS-STR instrument provided precise and accurate temperature and trace gas profiles (Woiwode et al., 2012). The GLORIA measurements aim for significantly higher vertical resolutions. Due to the higher spatial sampling along the flight track, a larger set of profiles is measured. Ungermann et al. (2015) discussed and validated GLORIA

temperature, $H_2O$, $HNO_3$ and $O_3$ retrievals from the GLORIA high spatial resolution mode. Woiwode et al. (2015) described the first GLORIA high spectral resolution observations captured in 2011 during the ESSenCe campaign. They compared temperature, $HNO_3$, $O_3$, $H_2O$, CFC-11 and CFC-12 profiles with in-situ profiles and MIPAS-STR collocated profiles, and they found that GLORIA, at this stage of development, showed reasonable agreement with MIPAS-STR and in-situ instruments. Here, a much higher number of profiles was measured, the instrument has been technically improved, $ClONO_2$ is presented as

additional trace gases and the results at flight altitude are compared to in-situ observations from the same platform HALO (High Altitude and LOng range aircraft). Additionally a first combination of GLORIA and Aura/MLS (Microwave Limb Sounder) data is presented.

The PGS mission is the combination of the POLSTRACC (POLar STRAtosphere in a Changing Climate) aircraft campaign (Oelhaf et al., 2015) together with GW-LCYCLE II (Gravity Wave Life Cycle Experiment) and SALSA (Seasonality of Air



mass transport and origin in the Lowermost Stratosphere using the HALO Aircraft) campaigns. The combined mission took place in the Arctic winter 2015/2016 with bases in Oberpfaffenhofen (Germany) and Kiruna (Sweden). The scientific objectives of the PGS campaign are among others to investigate chemical processes such as ozone depletion, chlorine de-/activation and de-/nitrification in the lowermost stratosphere as well as mixing and dynamical linkages between the upper troposphere, the

5  lower stratosphere and between high latitudes and middle latitudes over the course of the winter and gravity waves. For that purpose nine in-situ and three remote sensing instruments probed the UTLS region during 18 HALO research flights between December 2015 and March 2016. The flight paths are shown in Fig. 1. These 18 PGS research flights cover the whole time of the Arctic winter and provide with HALO flights of typically 10 hours of duration a unique data set.

The goal of the paper in hand is to characterize and validate the GLORIA observations during the course of the Arctic winter,

10  involving measurements under cloud-free conditions and conditions affected by polar stratospheric clouds (PSCs). The data product is characterized considering random and systematic errors, and an approach for correcting systematic line-of-sight errors in limb-imaging observations is presented. Finally, the GLORIA observations are brought into a broader perspective by comparisons with Aura/MLS observations and demonstrate the capability of GLORIA of resolving mesoscale structures in the UTLS. This paper shall provide the baseline and reference for scientific studies using GLORIA measurements.

## 2 Instruments

### 5 2.1 GLORIA

The essential parts of the GLORIA instrument (Friedl-Vallon et al., 2014, and the AMT special issue this paper belongs to) are an imaging spectrometer, a gimballed frame for pointing and line-of-sight stabilization and two black-bodies for radiometric calibration. The spectrometer is a Fourier Transform Spectrometer (FTS) with a HgCdTe infrared detector array. $48 \times 128$ (horizontal $\times$ vertical) pixels of this imaging detector are used to record the same number of interferograms simultaneously.

In order to reduce thermal noise, the spectrometer is cooled down to -50°C (Piesch et al., 2015). Depending on the scientific goals, the GLORIA spectrometer can be operated in two different measurement modes: The high spatial resolution mode with a spectral sampling of $0.625$ cm$^{-1}$ and a temporal resolution of 2 s and the high spectral resolution mode with a sampling of $0.0625$ cm$^{-1}$ and 13 s. In this paper results of measurements in high spectral resolution are discussed. In this measurement configuration $48 \times 128$ interferograms are recorded every 13 s, which corresponds to displacement of the carrier of $\approx$3 km

considering typical HALO cruise speed. The gimbal frame is used to compensate for the movements of the carrying aircraft and also offers the possibility to point at azimuth angles between 45° and 135° relative to the aircraft for measurements in across-track limb geometry. These different azimuth pointing angles are desired to avoid sun stray light, to correct for movements of the carrying aircraft due to cross winds or to adopt the measurement line of sight for expected horizontal gradients in temperature or trace gases. Another application of the adjustable azimuth angle is the feasibility of tomographic measurements

(Ungermann et al., 2011).

The level 1 processing comprises the generation of radiometrically and spectrally calibrated spectra from raw measurement data (Kleinert et al., 2014). At first, the interferograms are corrected for spikes and for the non-linearity of the detector and





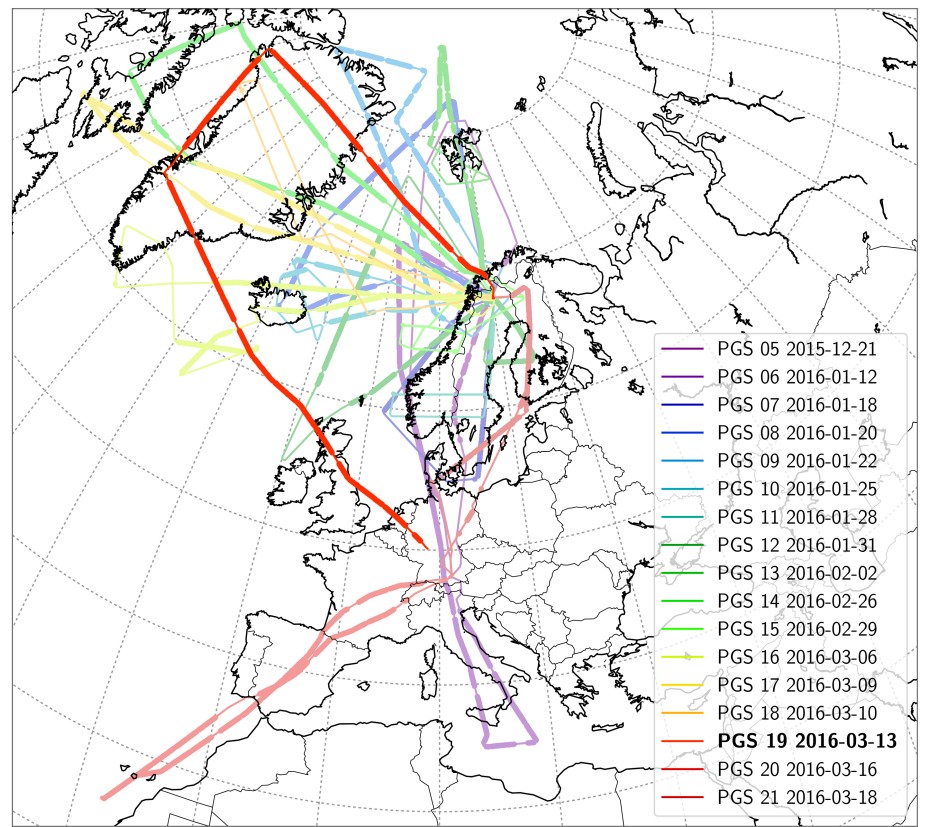

**Figure 1.** Flight paths of all PGS flights with GLORIA measurements. The parts of the flights with GLORIA high spectral resolution mode measurements are represented in bold lines. Flight PGS19 (13 March 2016) is discussed in detail in this paper and is highlighted in the map.

readout system. Then, they are re-sampled from the time-equidistant measurement grid onto a space-equidistant grid using information of a reference laser (Brault, 1996). During re-sampling, the interferograms are corrected for possible shifts due to

linear phase drifts, and the off-axis angle of each pixel is taken into account in order to sample each interferogram onto the correct abscissa in space. After the Fourier transform, a complex calibration according to Revercomb et al. (1988) is performed. Gain and offset are determined from regular in-flight measurements of the two on-board black-bodies at different temperatures. The spectra are apodized using the Norton-Beer "strong" apodization (Norton and Beer, 1976, 1977). This processing is done individually for each of the $48 \times 128$ interferograms. For noise reduction, the pixels of each detector row are averaged after

filtering of bad pixels. As measurements are smeared along-track due to the horizontal movement of the aircraft, this averaging does not result into a loss of information. This process results in 128 row-averaged spectra with different elevation angles. After cloud filtering, this set of spectra serves as input for the retrieval of atmospheric parameters. All atmospheric parameters are retrieved from the same set of averaged spectra.

GLORIA has been deployed during the HALO campaigns TACTS/ESMVal (2012), POLSTRACC/GW-LCYCLE/SALSA (2015/2016) and WISE (2017) and during the M55 Geophysica campaigns ESSenCe (2011) and StratoClim (2016/2017).





During these campaigns, the instrument has been constantly improved (Kretschmer et al., 2015), the data processing has been enhanced (Kleinert et al., 2014; Guggenmoser et al., 2015), the level 2 products have been validated (Kaufmann et al., 2015; Woiwode et al., 2015; Ungermann et al., 2015) and GLORIA data has proven to be useful for model validation (Khosrawi et al., 2017) and case studies (Rolf et al., 2015; Krisch et al., 2017). Improvements to the instrument (reduced aero-acoustic noise in the spectra) compared to the results of Woiwode et al. (2015) also increase the quality of the measured infrared spectra, resulting in different characteristics of the retrieved temperature and trace gas profiles.

## 2.2 In-situ instruments

On board of HALO several in-situ instruments were deployed during the PGS campaign. These in-situ instruments measure temperature and trace gases at the position of the aircraft with high precision and temporal resolution. Calibration measurements with reference gases or calibration units also allow a high accuracy of the measurements.

The AIrborne (chemical ionizaton) Mass Spectrometer (AIMS) measures $HCl$, $SO_2$, $HNO_3$ and $ClONO_2$ at a time resolution of 1.7 s with detection limits of 6-20 pptv and 10-15% precision with an accuracy of 12-20% (Jurkat et al., 2016, 2017). In addition, water vapor in low concentrations is measured in a second configuration (Kaufmann et al., 2016; Voigt et al., 2017).

Water vapor measurements between 1 and 1000 ppmv are performed with the Fast In-situ Stratospheric Hygrometer (FISH), which based on Lyman-$\alpha$ photo-fragment fluorescence (Zöger et al., 1999). FISH is one of the core airborne in-situ instruments for measuring water vapor in the UTLS (Fahey et al., 2014). FISH has a time resolution of 1 s and achieved a precision of 0.7% $\times$ vmr (volume mixing ratio) +/-0.35 ppmv with an overall accuracy of 6.6% $\times$ vmr during PGS (Meyer et al., 2015).

The Basic HALO Measurement and Data System (BAHAMAS) consists of a sensor package for basic meteorological parameters such as temperature, pressure, airflow, wind and humidity and a data acquisition system which provides additional interfaces into the aircraft avionic system and to an inertial reference system (Krautstrunk and Giez, 2012; Giez et al., 2017). Sensor data is available with a time resolution of 100 Hz, standard processing is based on a 10 Hz time resolution. The temperature measurement is based on an open wire resistance temperature sensor, which is contained in a special Total Air Temperature (TAT) inlet located in the nose section of the aircraft. These housings are heated to prevent ice formation and designed to separate droplets and particles from the probed airflow ahead of the sensor. The airflow is slowed down inside the housing in order to approach TAT via adiabatic heating. Data processing contains several corrections to account for deviations from ideal inlet behavior (Bange et al., 2013). These corrections limit the accuracy of the temperature determination to about 0.5 K, while the precision of the measurement is estimated to be about 0.03 K by means of auto-co-variance Function analysis.

The ozone detector FAIRO (Fast airborne ozone instrument) was deployed on HALO with a time resolution of 10 Hz (Zahn et al., 2012). The $O_3$ volume mixing ratio has a precision of $\approx$0.3 ppbv (at 10 Hz) and an uncertainty of $\approx$1.5%.

Additionally the Gas chromatograph for the Observation of Stratospheric Tracers Mass Spectrometer (GhOST-MS) provides measurements of CFC-12 in the electron capture detector channel (Obersteiner et al., 2016).





### 2.3 Aura/MLS

The NASA Earth Observing System Aura satellite was launched in July 2004 into a near-polar, sun-synchronous 705 km altitude orbit with the Microwave Limb Sounder (MLS) deployed on board. The Aura satellite flies in formation in the "A-Train" constellation of satellites and has an approximately 1:45 PM local equator crossing time. The Aura/MLS instrument is a successor to the MLS instrument on the Upper Atmosphere Research Satellite (UARS) and is a limb sounder analyzing the

thermal emission (wavelengths from 2.5 to 0.1 mm) of the atmosphere using seven radiometers to cover five spectral bands (Waters et al., 2006). The radiometers are pointing in the orbital flight direction and vertically scan the limb in the orbit plane approximately every 165 km. According to the orbit of the Aura spacecraft, the global coverage of measurements is from $82°$ S to $82°$ N. In this work, MLS version 4.2 (Livesey et al., 2017) $HNO_3$ and $O_3$ data are used. These data products have a vertical resolution of 3.0 - 4.5 km for $HNO_3$ and 2.5 - 3.5 km for $O_3$ and a horizontal resolution of 350 - 450 km and 300 -

550 km, respectively, in the UTLS. Both trace gas products have been validated for previous data versions (Santee et al., 2007; Froidevaux et al., 2008; Jiang et al., 2007).

### 2.4 ECMWF meteorological analysis

The input profiles for temperature, pressure and water vapor for GLORIA retrievals are taken from analysis data of the European Centre for Medium-range Weather Forecasts (ECMWF). These meteorological analyses from the "Atmospheric Model

high resolution" (HRES) are available every six hours with a horizontal resolution of 1 $°$ and 137 vertical levels up to a top pressure level of 0.1 hPa. The global fields of temperature, pressure and potential vorticity (PV) are interpolated on a vertical grid of absolute altitude.

### 3 Retrieval

In order to retrieve trace gas distributions from the calibrated spectral radiances, an inverse problem has to be solved. To this

end, we used the retrieval software KOPRAFIT (Höpfner, 2000), in which the forward radiative transfer is calculated by the radiative transfer model KOPRA (Karlsruhe Optimized and Precise Radiative transfer Algorithm, (Stiller, 2000)). KOPRA is a line-by-line radiation transfer model which is optimized for highly resolved spectral measurements. This software is used in the processing of MIPAS-Envisat, MIPAS-Balloon and MIPAS-STR limb measurements (von Clarmann et al., 2003; Wetzel et al., 2002; Woiwode et al., 2012). KOPRAFIT employs the Jacobians (derivatives of the radiance with respect to the fitted

atmospheric parameters) provided by KOPRA to fit the selected atmospheric parameters to the measured set of spectra. The inverse problem is solved by the Gauss-Newton iterative algorithm (Rodgers, 2000) with Tikhonov-Phillips regularization (Tikhonov and Arsenin, 1977; Phillips, 1962):

$$\mathbf{x}_{i+1} = \mathbf{x}_i + \left(\mathbf{K}_i^T \mathbf{S}_y^{-1} \mathbf{K}_i + \gamma \mathbf{L}^T \mathbf{L}\right)^{-1} \left(\mathbf{K}_i^T \mathbf{S}_y^{-1}(\mathbf{y} - \mathbf{f}(\mathbf{x}_i)) + \gamma \mathbf{L}^T \mathbf{L}(\mathbf{x}_a - \mathbf{x}_i)\right) \tag{1}$$

Here $i$ denotes the iteration index, $\mathbf{x}_i$ the vector containing the atmospheric state of step $i$, $\mathbf{y}$ the radiance measurement vector, $\mathbf{f}$

the radiative transfer function, $\mathbf{x}_a$ the a priori profile, $\mathbf{K}_i$ the Jacobian of $\mathbf{f}$ for $\mathbf{x}_i$, $\mathbf{S}_y$ the co-variance matrix of the measurement,



**L** the first order differential operator and $\gamma$ the regularization parameter. The regularization term $\gamma \mathbf{L}^T \mathbf{L}$ constrains the retrieval result to a smooth profile of the retrieved atmospheric quantity. In the applied formulation, the regularization avoids a bias to the retrieval result from an a-priori profile (Eriksson, 2000). The regularization parameters are chosen such that high vertical resolutions are obtained while unrealistic oscillations of the retrieved quantity are avoided.

The retrieval strategy in this work follows closely the one described by Woiwode et al. (2012). For the retrieval, the atmospheric parameters are represented at a discrete altitude grid with 250 m spacing in the region of interest (3 - 17 km) and coarser grid width below and above (1.5 km for 0 - 3 km, 2 km for 18 - 20 km, 2.5 km for 20 - 30 km and 50 km for 50 - 100 km). For the first step of the retrieval, trace gas profiles from the climatology by Remedios et al. (2007) are used for all important trace gases in the selected spectral range. Temperature, pressure and water vapor are taken from an interpolation of ECMWF analysis data to the GLORIA tangent points. For the water vapor retrieval, a constant profile of 10 ppmv is used as initial guess. The retrieval quantity is either the line of sight (LOS), temperature, vmr of $HNO_3$, $O_3$, $ClONO_2$ and CFC-12 or the logarithm of vmr of $H_2O$. To consider atmospheric aerosols and transparent clouds, the logarithm of an artificial continuum is part of the retrieval vector as it is described in Woiwode et al. (2015).

For the preparation of the retrieval, cloud-affected spectra are filtered. For that purpose the Cloud Index (CI) introduced by Spang et al. (2004) is calculated for each measured spectrum as the color ratio between the micro-windows 788.20 cm$^{-1}$ to 796.25 cm$^{-1}$ and 832.30 cm$^{-1}$ to 834.40 cm$^{-1}$. The CI is shown in Fig. 2 for the flights on 31 January 2016 and on 13 March 2016. Lower CI values indicate a larger influence of clouds on the spectrum. In previous studies using comparable airborne limb emission observations, typically fixed cloud index thresholds between 2 and 4 were used (Ungermann et al., 2012; Woiwode et al., 2012, 2015). In this work a CI threshold of 3 is used for the lowest and 1.8 for the highest limb tangent altitude to account also for observations moderately affected by PSCs. The CI thresholds for points in between are linearly interpolated. This approach is chosen to effectively filter out tropospheric clouds at lower altitudes while optically thin cirrus or polar stratospheric clouds still are allowed for the retrieval.

In the first step the line of sight is determined to correct a possible misalignment of the GLORIA gimbal frame. From this retrieval a correction for the line of sight is calculated and applied to all of the following steps. Then temperature, $HNO_3$, $O_3$, CFC-12, $ClONO_2$ and $H_2O$ are sequentially retrieved. After each step, the values retrieved for the previous quantities are kept fixed.

The spectral windows for the retrieval of the different trace gases are shown in Tab. 1. These spectral ranges were selected to minimize the cross-talk of emission lines of other trace gases, and saturation of spectral lines, particularly at low limb views, is minimized.

## 3.1 Error analysis

For the characterization of the results, possible error sources are estimated and their influences on the retrieval are calculated. In this work, we estimate systematic and noise errors. Considered errors are spectroscopic uncertainties (as reported in previous work), radiometric calibration errors (multiplicative gain and additive offset), residual pointing uncertainty and a temperature error for vmr retrievals and radiometric calibration errors, pointing uncertainty and an error of the $CO_2$ climatology profile



**Table 1.** Spectral windows for the different target species of the GLORIA high spectral resolution mode PGS retrieval

| Retrieval target | Micro-window [$cm^{-1}$] |
| --- | --- |
| LOS and Temperature | 810.5 - 812.9 |
| | 956.0 - 958.2 |
| $HNO_3$ | 862.0 - 863.5 |
| | 866.1 - 867.5 |
| | 901.3 - 901.8 |
| $O_3$ | 780.6 - 781.7 |
| | 787.0 - 787.6 |
| $ClONO_2$ | 780.0 - 780.4 |
| $H_2O$ | 795.7 - 796.1 |
| CFC-12 | 918.9 - 921.3 |

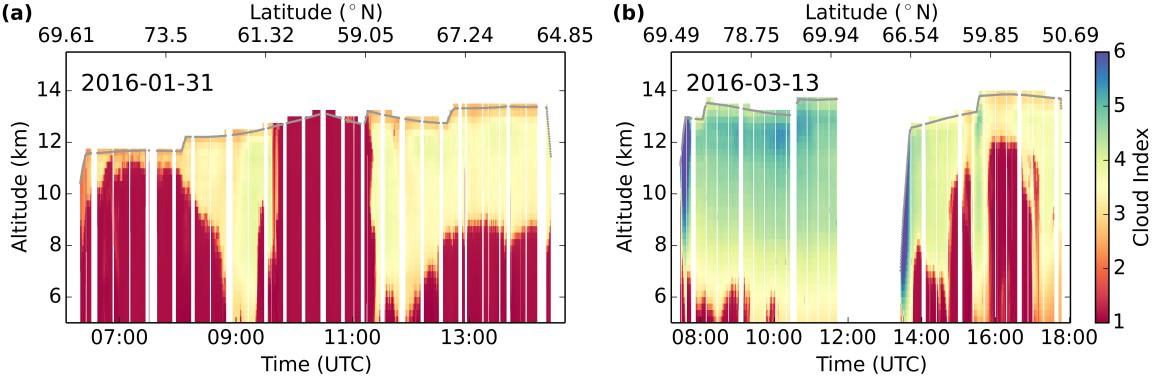

**Figure 2.** Vertical distribution of cloud indices along flights PGS12 (a, 31 January 2016) and PGS19 (b, 13 March 2016). The lower the cloud index, the more is the measured spectrum affected by clouds. Between 12:00 and 14:00 UTC in PGS19 no atmospheric data was measured due to a refueling stop of the aircraft.

(accounting also for errors in the $CO_2$ spectroscopic line data).

The spectroscopic error is estimated as 8% for $HNO_3$ (Wetzel et al., 2002), 5.5% for $ClONO_2$ (Wagner and Birk, 2003) and 10% for CFC-12 (Moore et al., 2006). For $O_3$ and $H_2O$ uncertainties in line intensities are reported (Flaud et al., 2002, 2006) and the spectroscopic error is estimated as 7% for $O_3$ and 10% for $H_2O$. Considering the temperature retrieval, the spectroscopic error can be estimated by assuming an error in the $CO_2$ profiles as high as 5.0 %, according to Wetzel et al. (2002).




In order to quantify the influence of the assumed $CO_2$ profile, the temperature retrieval has been repeated with a $CO_2$ profile uniformly decreased by 5%. The differences between these retrievals at each grid point show the sensitivity of the retrieval to the modified $CO_2$ profile.

In the same way the impacts of further error sources on the retrieval are quantified: The results of a retrieval with a perturbed radiometric calibration, LOS or temperature are subtracted from the standard retrieval to estimate the error of each individual retrieval grid point. With this method uncertainties in the radiometric calibration, which can be attributed to errors in the multiplicative gain and the additive radiance offset, are calculated considering uncertainties in multiplicative gain enhancement of 2% and an additive 50.0 $nWcm^{-2}sr^{-1}cm$ radiance offset. LOS errors are estimated by retrievals involving a 0.05° LOS offset

considering the profile-to-profile variability found in the LOS retrieval and systematic uncertainties inherent to the LOS retrieval. For trace gas retrievals, the retrieved temperature is used to describe the atmospheric state. The effect of uncertainties in the retrieved temperatures on the trace gas retrievals is estimated by modifying the retrieved temperature profile systematically with the related temperature error. The retrieval noise is calculated according to Rodgers (2000)

$$\Delta x_{\mathrm{noise}} = \mathbf{G}_y \epsilon = \left( \left( \mathbf{K}^T \mathbf{S}_y^{-1} \mathbf{K} + \gamma \mathbf{L}^T \mathbf{L} \right)^{-1} \mathbf{K}^T \mathbf{S}_y^{-1} \right) \cdot \epsilon \,. \tag{2}$$

Here $\Delta x_{\mathrm{noise}}$ denotes the noise error, $\mathbf{G}_y$ the retrieval gain matrix, $\epsilon$ the measurement error and $\mathbf{K}$ the Jacobian for the last iteration step. This measurement error is estimated as the spectral variance in the micro-window of the imaginary part of the calibrated spectrum (Kleinert et al., 2014).

The total estimated error for each altitude of each retrieved profile is calculated as the square root of the sum of the squares of each error contribution, as is shown for vmr in Eq. (3) and for temperature in Eq. (4).

$$\Delta x_{\mathrm{vmr}} = \sqrt{\Delta x_{\mathrm{spectroscopy}}^2 + \Delta x_{\mathrm{gain}}^2 + \Delta x_{\mathrm{offset}}^2 + \Delta x_{\mathrm{pointing}}^2 + \Delta x_{\mathrm{temperature}}^2 + \Delta x_{\mathrm{noise}}^2} \tag{3}$$

$$\Delta x_{\mathrm{temperature}} = \sqrt{\Delta x_{CO_2}^2 + \Delta x_{\mathrm{gain}}^2 + \Delta x_{\mathrm{offset}}^2 + \Delta x_{\mathrm{pointing}}^2 + \Delta x_{\mathrm{noise}}^2} \tag{4}$$

### 3.2   Vertical resolution and degrees of freedom

An important diagnostic measure is the vertical resolution, which is calculated by using the averaging kernel of the retrieval. The averaging kernel matrix is defined as (Rodgers, 2000)

$$\mathbf{A} = \mathbf{G}_y \cdot \mathbf{K} \,. \tag{5}$$

The vertical resolution at a retrieval grid point is calculated as the full width at half maximum of the averaging kernel row. Another important quantity for a retrieval is the degrees of freedom. These are calculated as the trace of the averaging kernel matrix (Rodgers, 2000), since the diagonal element of each averaging kernel row is a measure of how much measurement information is contained in the retrieval result.



## 4   Results

The results of the GLORIA measurements for the flight on 13 March 2016 (PGS19) are shown in the following part. Results
for all of the other 14 PGS research flights with GLORIA measurements in high spectral resolution are shown in the appendix.
First, the meteorological and chemical background situation of this flight day and region is discussed on the basis of MLS

measurements at a level corresponding with a typical flight altitude for this specific flight. Then one example temperature and
HNO$_3$ profile is characterized in detail. The main part of this section is the discussion of the GLORIA retrieval results for flight
PGS19. For the discussion of the elevation angle correction also results for the flight on 31 January 2016 (PGS12) are shown
as an example for a different type of LOS distortion and correction. In order to provide a survey of all GLORIA measurements
and their quality during the whole PGS campaign, comparisons to in-situ and MLS measurements are presented as an overview.

### 4.1   Meteorological situation for flight PGS19 on 13 March 2016

The flight PGS19 on 13 March 2016 was the transfer back from the campaign base in Kiruna, Sweden to Oberpfaffenhofen,
Germany. This flight was planned to sample aged vortex air over the northwestern part of Greenland and to cross a region of
subtropical air associated with a high tropopause between the East coast of Greenland and Ireland. Take-off in Kiruna was at
07:08 UTC and touch-down in Oberpfaffenhofen at 18:28 UTC. The first part of the flight was directed towards the north-

western coast of Greenland (see Fig. 3 way point "A"), flying over the Norwegian sea and then in a southern direction to a
refueling stop at Kangerlussuaq airport in Greenland (way point "B"). This stop allowed for higher altitudes of the HALO
aircraft before refueling due to lower aircraft weight. Otherwise, the necessary altitude to sample subsided polar air masses
over the northern part of Greenland could not have been reached. After this stop, the aircraft passed the northern Atlantic ocean
and the British Isles towards Oberpfaffenhofen in southern Germany.

The meteorological situation during this flight is shown in Fig. 3 (top row) with temperature (a) and potential vorticity (PV, b)
at a level corresponding to a typical flight altitude of 13 km for this specific flight. At this altitude and time of the winter, the
PV determining the edge of the polar vortex according to Nash et al. (1996) is estimated as $\approx$ 9 PVU. The first part of the flight
(until way point "A") took place in relatively warm 220 to 230 K air masses compared to the rest of the flight. PV increased
along the flight track towards maximum values of more than 12 PVU. This indicates that the flight entered the late winter polar

stratosphere and presumably even aged subsided polar vortex air. During the flight leg between the way points "A" and "B", the
aircraft remained within these stratospheric air masses with relatively warm temperatures and high PV. On the flight leg from
way point "B" towards the final destination Oberpfaffenhofen, HALO departed these air masses, with temperatures decreasing
down to 200 K and PV down to 4 PVU over the northern Atlantic ocean. The air masses above the British Isles and central
Europe showed temperatures of up to 210 K and PV of 6 to 9 PVU with fine filaments visible on the PV map. This might point

to air masses remaining from the dissolving late winter polar vortex.

For a comparison of the MLS measurements with GLORIA, the MLS data has been selected for days of and time periods around
PGS flights, filtered regarding data quality as recommended by Livesey et al. (2017) and interpolated onto a regular horizontal
grid (2° latitude × 4° longitude) using a squared cosine as the weighting function. The width of this squared cosine function



has been chosen to be 1.5° for latitudes and 8.0° for longitudes, and a minimum threshold of 0.75 is selected. Additionally, the pressure coordinate of the MLS data is interpolated to geometric altitude for an easier comparison to the GLORIA data. This interpolation method does not provide meaningful comparisons of water vapor because tropospheric $H_2O$ (in contrast to stratospheric $HNO_3$ and $O_3$) is likely to significantly change within the time range of Aura/MLS measured profiles, which are selected for this type of interpolation. For this reason no comparisons of GLORIA and Aura/MLS $H_2O$ measurements are shown.

An overview of these gridded MLS $HNO_3$ and $O_3$ horizontal distributions is shown in Fig. 3 (bottom row) at a typical flight altitude of 13.0 km for PGS flight 19 on 13 March 2016. The HALO flight track and geolocations (position and altitude) of GLORIA tangent points are also shown on these maps. Along the flight track local minima in MLS $HNO_3$ are observed above the Norwegian sea and south of Iceland. Local maxima in MLS $HNO_3$ and $O_3$ are present above the northern part of Greenland at way point "A".

## 4.2 Characterization of example profiles

The quality of the retrieved GLORIA data can be assessed by the estimated errors (see Sect. 3.1) and by the vertical resolution (see Sect. 3.2). These quantities are shown in Fig. 4 as an example for the retrieval result of a temperature and nitric acid profile. In the left column (a,d) of this figure the retrieval results and the initial guess profiles are shown; the right panel (c,f) shows the vertical resolution. For these retrievals vertical resolutions of 400 m to 750 m are achieved. It can be seen that the retrieved temperature and vmr profiles and their shapes significantly differ from the initial guess profiles at an altitude range between 5.5 km and 13.5 km, reflecting the weak influence by the Tikhonov regularization. Below these altitude ranges the profile shapes resemble those of the initial guess profile and no information is contributed by the measurement. Above 13.5 km, the retrieved profiles are also influenced by measurements with upward looking lines of sight, which explains the small differences in shape at these altitudes. In this region the vertical resolution is poor and thus little measurement information is obtained for these parts of the profiles.

In the second column (b,e) of Fig. 4 individual $1\sigma$ error contributions are shown. For the temperature the total estimated error is most influenced by radiometric gain calibration (up to 1.0 K) and pointing uncertainties (in the range of 0.5 to 1.5 K). In the case of $HNO_3$ the total estimated error is dominated by the spectroscopy error estimated as a constant relative fraction of 8.0 % (as assumed in Sec. 3.1) and the uncertainties due to the previously retrieved temperature data (up to 0.3 ppbv). The pointing error has large contributions (up to 0.6 ppbv) to the total estimated error at altitude ranges where large vertical gradients in the profiles occur. At these altitude ranges with large vertical gradients even small changes in the elevation pointing have large influence on the absolute differences between the perturbed and the reference retrieval result. The radiometric offset and the retrieval noise error only contribute a minor part of the total error ($\leq$0.5 K for temperature and up to $\leq$0.2 ppbv for $HNO_3$).

In the following discussion of the results, the retrieved profile, vertical resolution, total estimated error are presented as curtain plots for a whole research flight. This shall provide an overview of the amount of data that has been measured by GLORIA and still allow the characteristics of the results to be shown in detail.





**Figure 3.** (a) Temperature and (b) potential vorticity (PV) from ECMWF meteorological analysis on 13 March 2016 12:00 UTC at 13.0 km and (c) MLS HNO$_3$ and (d) O$_3$ measurements on 13 March 2016 between 6:00 and 18:00 UTC (approximation for the time period of flight PGS19) interpolated to a regular latitude/longitude grid and typical HALO cruising altitude of 13.0 km. For flight PGS19 on this day, the ground track of the HALO aircraft is shown with a magenta line and the geolocations of GLORIA tangent points are shown along the flight track with points in the greyscale color map. Way points of this flight are marked with capital letters.





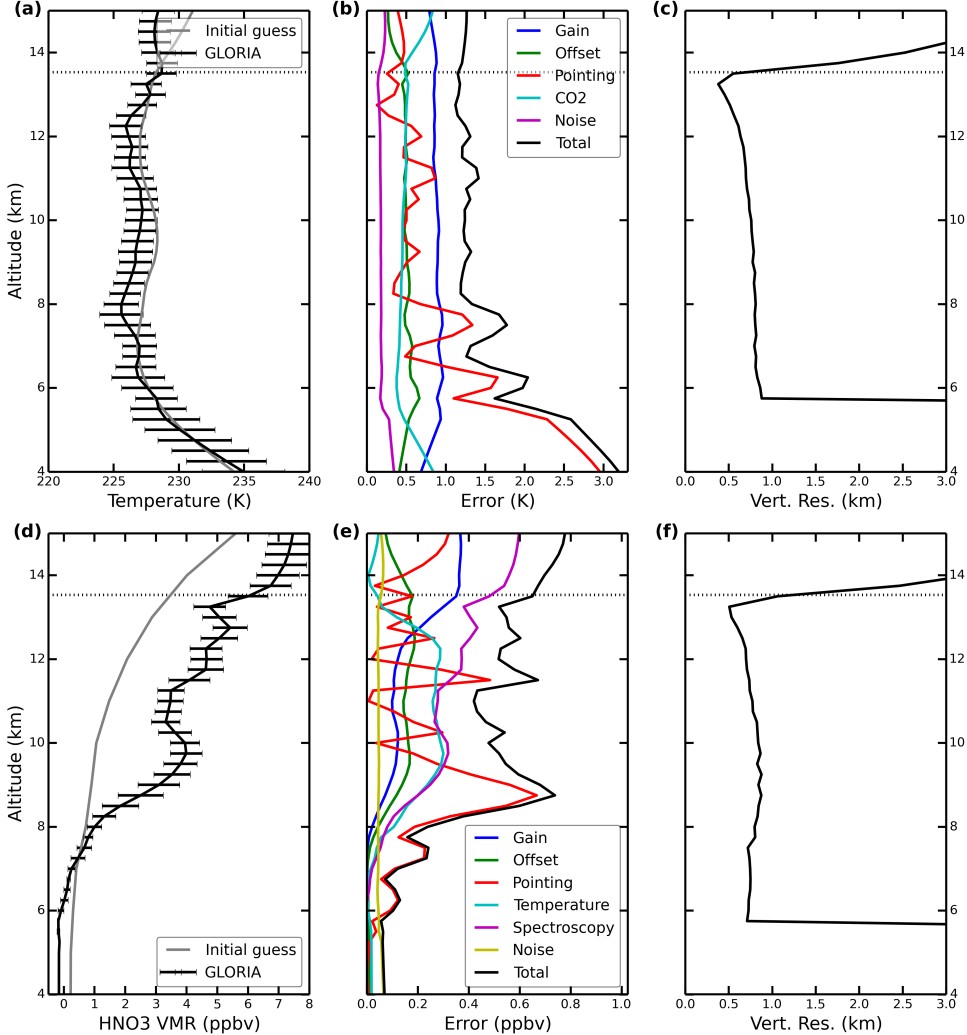

**Figure 4.** Illustration of the error budget of the temperature and HNO₃ retrieval of Flight 19 on 13 March 2016 for a selected profile at 10:37:06 UTC. First column: retrieved vertical profile of temperature (first row) and HNO₃ (second row) with total estimated error (black) and initial guess profile (grey). The retrieved profile has 14.9 and 12.5 degrees of freedom for temperature and HNO₃, respectively. Second column: Different total error contributions and estimated total error. Last column: Vertical resolution of this retrieval result. The dotted line represents the flight altitude of the aircraft.

## 4.3 GLORIA results

### 4.3.1 Line of sight

For each flight, one systematic pointing correction is derived from the retrieval. The pointing elevation angle is retrieved to compensate for systematic misalignment of the pointing system of GLORIA. The LOS retrieval results for the flights on 31





January 2016 and on 13 March 2016 are shown as the difference between expected and retrieved elevation angle (black dots in Fig. 5). This difference in the retrieved LOS can be caused by changes of the atmospheric state compared to the ECMWF fields, thermal deformation of the instrument, and a systematic error of the pointing calibration on ground. Fluctuations in the retrieved LOS are attributed to the changes of the atmospheric state not resolved or reproduced by the ECMWF fields and the thermal deformation, while the systematic error of the calibration is expected to result in a time-independent LOS offset. With the diagnostic data available, it is not possible to distinguish between atmospheric variations and thermal deformations of the instrument. Ground-based measurements suggest that the thermal deformations of the instrument are an important cause for these variations, but also the atmospheric variation of temperature and pressure is estimated to have a major impact. For that reason generally, only one average LOS correction value per flight is used. For future campaigns it is planned to ensure the

quality of the pointing by ground-based absolute pointing calibrations and by in-flight measurements of the moon on a regular basis. An example of this average correction, which is applied for flights between 02 February 2016 and 18 March 2016, is shown in Fig. 5 (b, blue points).

For flights between 21 December 2015 and 31 January 2016 a software malfunction of the pointing control software caused the LOS to drift away from the commanded elevation. At certain points the software changed the instrument elevation to its

correct value and steep steps in the retrieved pointing elevation angle are observed in these flights (see Fig. 5 a). A correction of this artifact can be calculated by interpolating the LOS between the points immediately after a steep step. This interpolated line approximates the LOS that would have been retrieved for a measurement without this software malfunction. The same average LOS correction, which is used for other flights, can be calculated from this interpolated LOS (Fig. 5 a, green points). The influence of the software malfunction can be extracted by subtraction of the interpolated LOS from the retrieved LOS. For

subsequent retrievals of temperature and volume mixing ratios, both corrections, the average LOS correction and the correction of the steps, have been applied (Fig. 5 a, blue points).

### 4.3.2  Temperature

The retrieved temperature along with characterization diagnostics and comparison to in-situ observations for the flight PGS19

is shown in Fig. 6. The retrieval result in the panel (a) shows the temperature profiles in a color coded curtain plot. This type of plot is also used to present the curtains of volume mixing ratio results. The lower horizontal axis indicates the measurement time and the upper horizontal axis the corresponding latitude of the aircraft. The vertical axis shows the absolute altitude of the retrieval grid points. The retrieval result is filtered according to the vertical resolution. Only data points with a vertical resolution better than 2 km are presented. For that reason the data above flight level is filtered out. Also the measured spectra

below cloud tops have been filtered out prior to the retrieval. The time between 12:00 and 13:00 UTC was spent on the ground due to a refueling stop of the aircraft. Smaller gaps between the profiles are due to radiometric calibration measurements. As a measure for the dynamical tropopause, the ECMWF potential vorticity interpolated to the GLORIA tangent points is shown in magenta dashed lines, marking the values of 2.0 and 4.0 PVU. In the first part of the flight (until way point "B"), high temperatures are observed. The dynamical tropopause is at low altitudes down to 6 km, and mainly stratospheric air masses





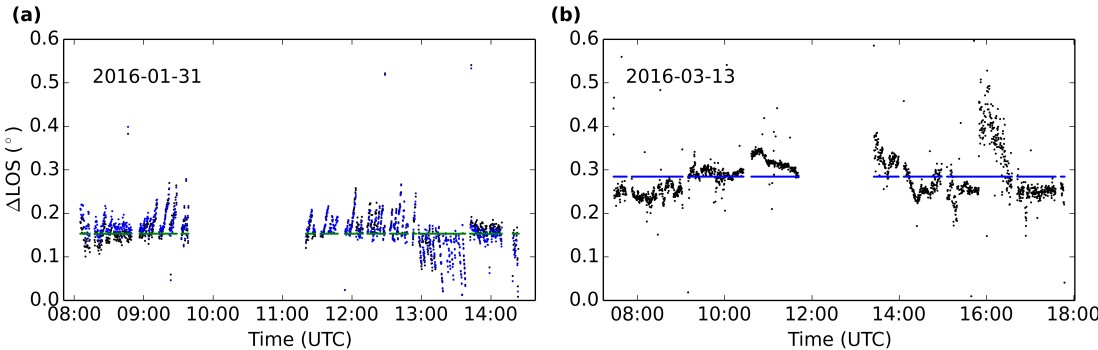

**Figure 5.** Line of sight correction of two flights (a: PGS12, b: PGS19) during the PGS campaign. The retrieved deviation of the LOS from the nominally set value is shown in black, the applied correction in blue and for flight PGS12 the averaged LOS (before applying the correction of the drift) in green.

are sampled by GLORIA during this part of the flight. These stratospheric air masses at low altitudes suggest subsidence of air masses from the polar vortex during the late winter. The second part of this flight shows the transition to a higher tropopause up to 12 km and also stronger vertical gradients from higher temperatures (240 K) at lower altitudes to lower temperatures down to 205 K at flight altitude. The last hour of measurements shows again a lower tropopause and less steep vertical gradients.

The total estimated error (b) indicates for most data points values in the range of 1.0 to 1.3 K. Especially at regions with higher temperature the retrieval results are less accurate due to higher gain error contributions. The main error contribution is the pointing error due to vertical gradients. The vertical resolution (c) of the temperature retrieval is between 500 m and 800 m. Altitudes closer to the aircraft usually show a better vertical resolution due to denser spacing of the tangent points.

In-situ measurements taken at flight level are compared to the GLORIA retrieval results obtained close to the flight altitude. From each vertical profile retrieved from GLORIA the grid point which is closest to the flight altitude (i.e. between 0 and 250 m underneath the flight altitude) and which has a vertical resolution better than 2 km is chosen for comparison. This assures the best possible match of sampled air masses with the in-situ instrument. It is important to keep in mind, that the data sets do not probe exactly the same air-masses, since GLORIA measures at the limb and thus collects the radiation from a long path of $\approx$ 100 km through the atmosphere (Ungermann et al., 2012, 2011). In Fig. 6(d) the comparison of GLORIA temperatures (green dots) to the BAHAMAS in-situ measurements (blue dots) is presented. The two measurements show agreement to within 1.3 K, which is the estimated error of the GLORIA temperature retrieval.

### 4.3.3 Nitric acid

Due to the formation and sedimentation of polar stratospheric clouds and the resulting de- or re-nitrification (Peter and Grooß, 2011), nitric acid ($HNO_3$) is expected to display irregular small structures in the UTLS region. For that reason, it is important to achieve spatially highly resolved and validated measurements of $HNO_3$ in the UTLS. The GLORIA retrieval results for the





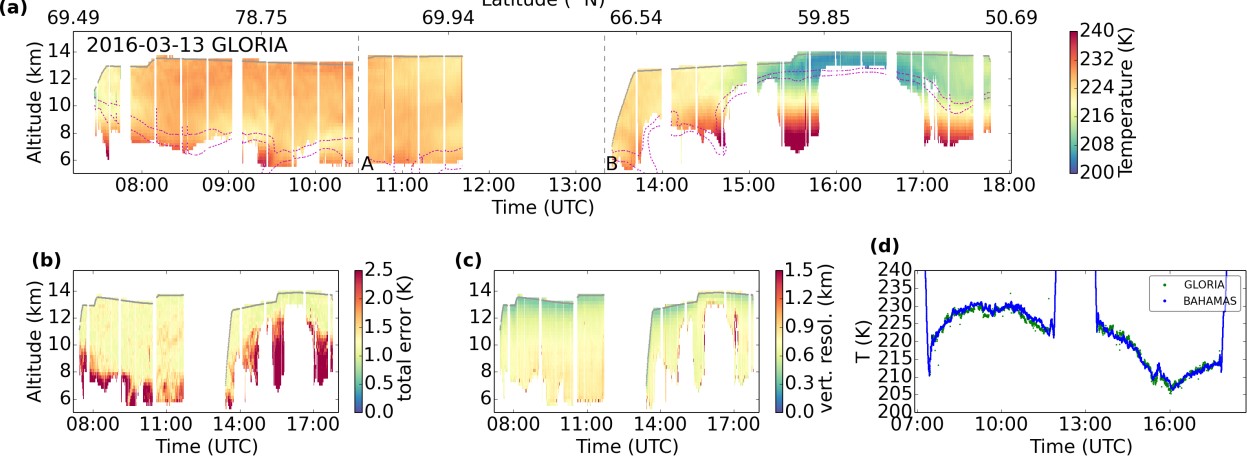

**Figure 6.** Temperature flight PGS19: Cross section of (a) retrieved temperature (the flight altitude is marked with a grey line; white spaces mark regions without data, the ECMWF potential vorticities of 2 and 4 PVU are marked with magenta dashed lines, way points are marked with grey vertical dashed lines) and cross sections of (b) estimated total error and (c) vertical resolution, followed by (d) comparison of the GLORIA measurements (green) to the BAHAMAS in-situ measurement (blue).

flight PGS19 are presented in Fig. 7. The two-dimensional distribution of $HNO_3$ volume mixing ratios shows fine structures with maximum values up to 7 ppbv. The retrieval has a typical vertical resolution of 500 to 800 m, and the error is typically 0.5

15  ppbv. The comparison to the in-situ measurements by AIMS is given in Fig. 7(e). The strong fluctuations of $HNO_3$ are captured simultaneously by both instruments. The agreement between the instruments is often better than 0.5 ppbv. However, at some locations their differences reach up to 2.0 ppbv. These discrepancies reflect the large atmospheric variability likewise in the horizontal direction due to de-/nitrification processes along the GLORIA line of sight. The horizontal distribution of PV (Fig. 3(b)) suggests that at this part of the flight (from way point "B" to the final destination Oberpfaffenhofen) air masses influenced

by outflow of the polar vortex are sampled, which explains higher variability in trace gas distributions. This atmospheric variability is also implied in the MLS $HNO_3$ horizontal distribution along the GLORIA viewing direction as shown in Fig. 3(c). For a qualitative comparison to the GLORIA measurements, the gridded MLS $HNO_3$ data has been interpolated to the GLORIA tangent points (Fig. 7(b)). Considering the different spatial resolutions of the GLORIA and the MLS data, both $HNO_3$ distributions show relative minima and maxima at the same locations and the absolute values are of the same order of

magnitude. Due to the lower vertical resolution of MLS $HNO_3$ measurements, they are more influenced by air masses at higher altitudes and small structures cannot be resolved. This difference in spatial resolution explains lower absolute $HNO_3$ in MLS compared to GLORIA. The advantage of the satellite product, though, is information about air masses above the HALO flight altitude and how these large scale structures of $HNO_3$ are connected with the filaments measured by GLORIA.



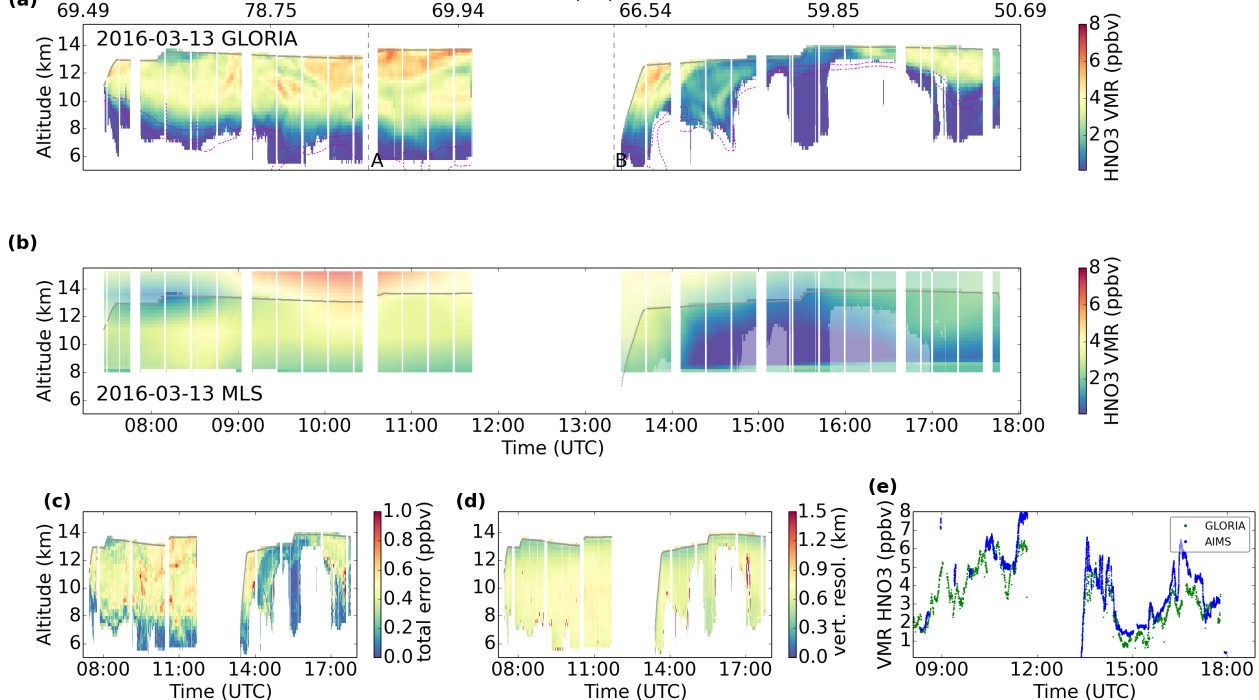

**Figure 7.** $HNO_3$ flight PGS19: Cross section of (a) retrieved $HNO_3$ volume mixing ratio (the flight altitude is marked with a grey line, the ECMWF potential vorticities of 2 and 4 PVU are marked with magenta dashed lines, way points are marked with grey vertical dashed lines). Cross section of (b) MLS $HNO_3$ data interpolated to the GLORIA tangent points and above the aircraft. Regions with no corresponding GLORIA measurement are marked by fainter colors. Cross sections of (c) total estimated error, (d) vertical resolution and (e) comparison of the GLORIA measurements (green) to the AIMS in-situ measurement (blue)

### 4.3.4 Ozone

The measured ozone ($O_3$) distribution during the flight PGS19 can be found in Fig. 8, where maximum values up to 1600 ppbv at altitudes of 13 km are observed. Below this maximum, finer structures are present. The total estimated error (up to 150 ppbv) is dominated by spectroscopic and gain uncertainties. Vertical resolutions from 500 to 900 m are achieved. In comparison to the FAIRO in-situ measurements, the GLORIA retrieval results follow the long-term as well as the short-term variations. The agreement of the two measurements is typically better than 100 ppbv. In regions of maximum observed $O_3$ mixing ratios high profile-to-profile variations up to 200 ppbv are visible. These variations are explained by the estimated total error and by the expected atmospheric variability along the GLORIA line of sight. In the second part of the flight, GLORIA and FAIRO ozone data show different structures and differences between the measurements up to 300 ppbv. This is the same region, where the

5 $HNO_3$ in-situ comparison shows differences and where an inhomogeneous horizontal distribution is suspected to distort the comparison of these measurements. The horizontal distribution of $O_3$ at 13 km altitude as derived from MLS measurements is





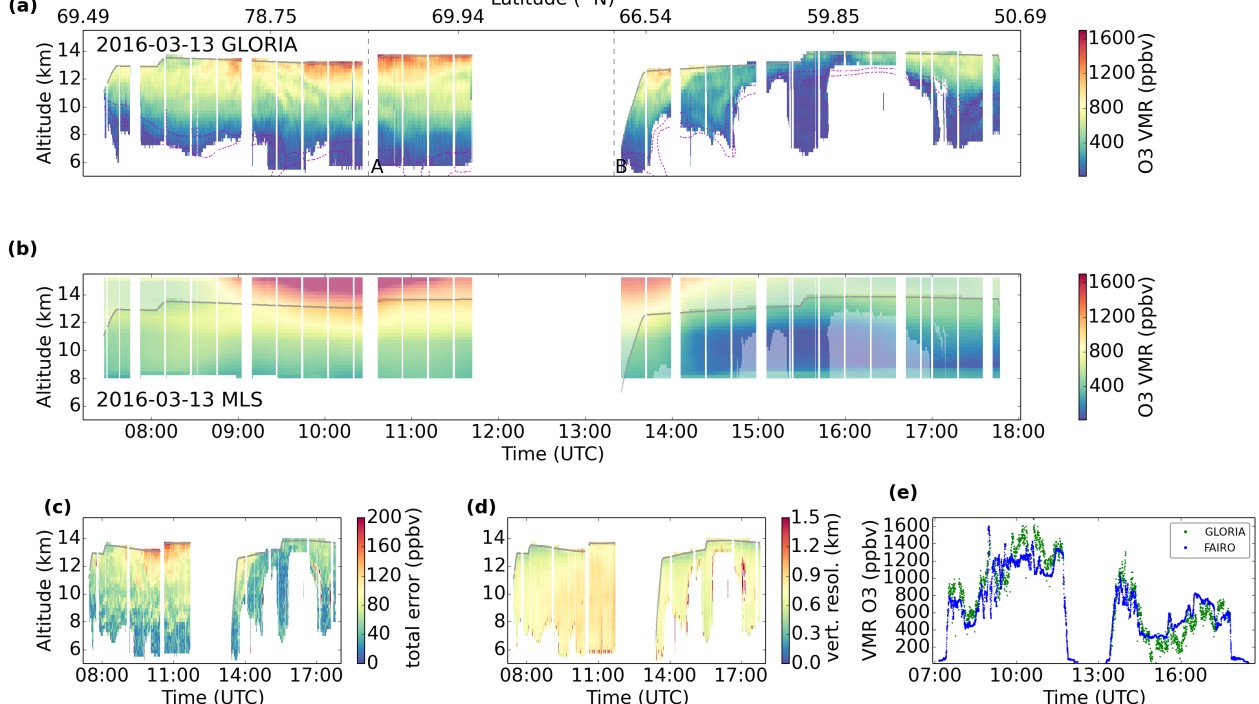

**Figure 8.** $O_3$ flight PGS19: Cross section of (a) retrieved $O_3$ volume mixing ratio (the flight altitude is marked with a grey line, the ECMWF potential vorticities of 2 and 4 PVU are marked with magenta dashed lines, way points are marked with grey vertical dashed lines). Cross section of (b) MLS $O_3$ data interpolated to the GLORIA tangent points and above the aircraft. Regions with no corresponding GLORIA measurement are marked by fainter colors. Cross sections of (c) total estimated error, (d) vertical resolution and (e) comparison of the GLORIA measurements (green) to the FAIRO in-situ measurement (blue).

illustrated in Fig. 3d. A horizontal gradient is seen above Baffin Bay (higher $O_3$ volume mixing ratios in the GLORIA line of sight compared to the aircraft position). The comparison of GLORIA $O_3$ to the MLS distributions interpolated to the GLORIA geolocations shows very similar large-scale structures in both data sets. Again, small scale structures which are visible in GLORIA measurements are not captured in the lower resolution MLS data.

### 4.3.5 Chlorine nitrate

Chlorine nitrate ($ClONO_2$) is one of the two reservoir species (the other being HCl) of chlorine in the stratosphere. As was initially shown by infrared limb emission observations (Clarmann et al., 1993; Oelhaf et al., 1994; Roche et al., 1994) chlorine 5 deactivation in the Arctic spring region results in strong enhancement of $ClONO_2$. The retrieved $ClONO_2$ distribution in Fig. 9 shows several fine structures and maximum values up to 1500 pptv in the flight section (way points "A" to "B") that reached the highest PV values (see Fig. 3(b)), which can be interpreted as subsided deactivated $ClONO_2$. The corresponding values of





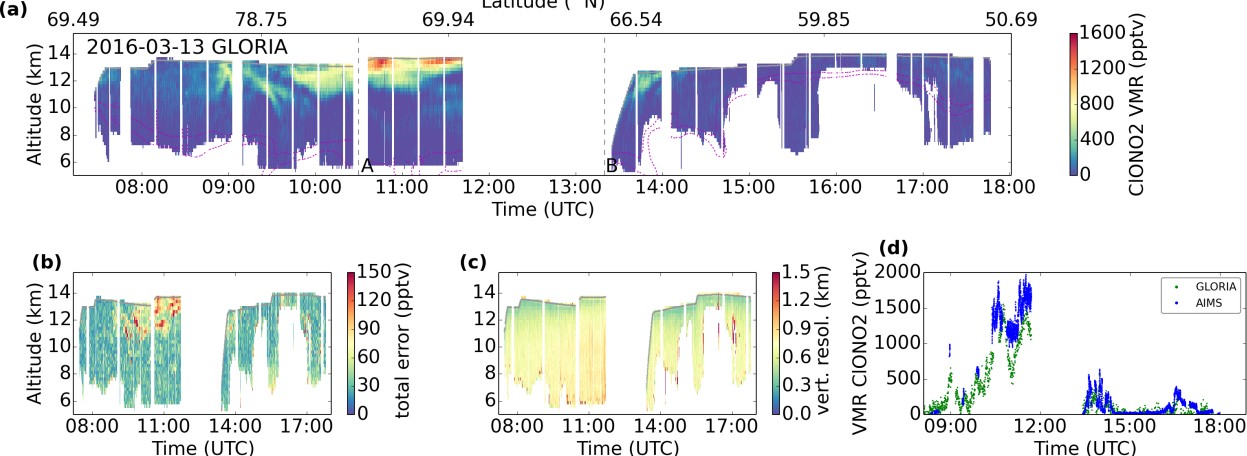

**Figure 9.** ClONO$_2$ flight PGS19: Cross section of (a) retrieved ClONO$_2$ volume mixing ratio (the flight altitude is marked with a grey line, the ECMWF potential vorticities of 2 and 4 PVU are marked with magenta dashed lines, way points are marked with grey vertical dashed lines). (b) Cross sections of total estimated error, (c) vertical resolution and (d) in-situ comparison of the GLORIA measurements (green) to the AIMS in-situ measurement (blue).

total estimated error are 150 pptv. In case of background values (< 100 pptv), the estimated errors are 30 pptv. The increased errors for enhanced values are caused by the impact of relatively increased gain errors. Vertical resolutions of 500 m to 900 m

10 are calculated for this retrieval. The comparison to the AIMS in-situ measurements of ClONO$_2$ shows agreement of the two data products to within 200 pptv, except for the maximum values, where differences are up to 500 pptv. Here, the GLORIA retrieval shows a lower absolute vmr but a similar structure. Again, we attribute this difference to an offset in altitude and a possible horizontal gradient, which is measured as an average along the GLORIA line of sight.

### 4.3.6 Water vapor

5 Water vapor (H$_2$O) is mainly present in the troposphere. GLORIA H$_2$O distributions are interesting for investigations of mesoscale structures such as tropopause folds (Shapiro, 1980). In polar studies, H$_2$O is of interest because these distributions are used to understand the formation and decay of polar stratospheric ice clouds (de-/hydration) (Fahey et al., 1990). The distribution of H$_2$O reflects the tropopause altitude with very low stratospheric values ($\approx$ 5 ppmv) in the region of the aged vortex (way points "A" to "B") and high values (10-20 ppmv) above the intrusion of subtropical air where HALO was close

10 to the tropopause (way point "B" to the final destination Oberpfaffenhofen). The total estimated error shows higher values (> 5 ppmv) in regions with enhanced H$_2$O vmr compared to errors lower than 1 ppmv in regions with measured stratospheric background values. The vertical resolution is between 400 m and 700 m. The comparison to the FISH in-situ measurement shows agreement to within the GLORIA error of typically 1 ppmv. The enhancement at flight altitude at 15:30 UTC is well captured in both data sets.





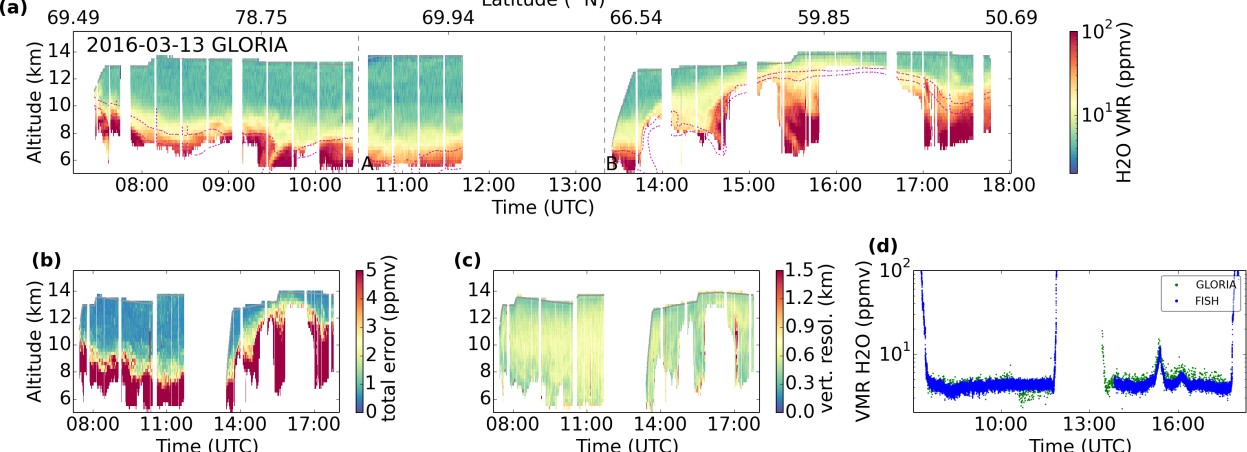

**Figure 10.** $H_2O$ flight PGS19: Cross section of (a) retrieved $H_2O$ volume mixing ratio (the flight altitude is marked with a grey line, the ECMWF potential vorticities of 2 and 4 PVU are marked with magenta dashed lines, way points are marked with grey vertical dashed lines). Cross sections of (b) total estimated error,(c) vertical resolution and (d) comparison of the GLORIA measurements (green) to the FISH in-situ measurement (blue).

### 4.3.7 Chlorofluorocarbon 12

Dichlorodifluoromethane (CFC-12) is a chlorofluorocarbon that has been artificially produced for usage as refrigerants and aerosols. Its production is regulated by the Montreal Protocol due to its potential for ozone depletion (WMO, 2015). Because of its vertical gradient, CFC-12 can be used as a tracer for tropospheric air and for the altitude of the air masses (Greenblatt

5  et al., 2002). The vmr distribution along flight PGS19 of CFC-12 is presented in Fig. 11. Here, mainly volume mixing ratios of about 500 pptv are observed in the troposphere. In the area where aged subsided vortex air was reached (way points "A" to "B") values as low as 320 pptv were found. The error is between 40 and 130 pptv. The vertical resolution is in the range of 500 m to 1000 m. The comparison to the in-situ measurements by GhOST-MS shows agreement to within 70 pptv. The high profile-to-profile variation up to 100 pptv of GLORIA CFC-12 that can be seen in this in-situ comparison plot exceeds the total

10  estimated error at flight altitude of ≈70 pptv. So it is likely, that atmospheric variability along the GLORIA line of sight might cause these fluctuations. This variability is also present in all other flights of this campaign (see Tab.2). Also, compared to other GLORIA retrievals, a higher number of extreme outlier points are observed in the GLORIA data. This is an indication that the retrieval for CFC-12 is more sensitive to perturbations in the spectra (e.g. high altitude clouds that have not been effectively filtered) compared to the retrievals of temperature and other trace gases.



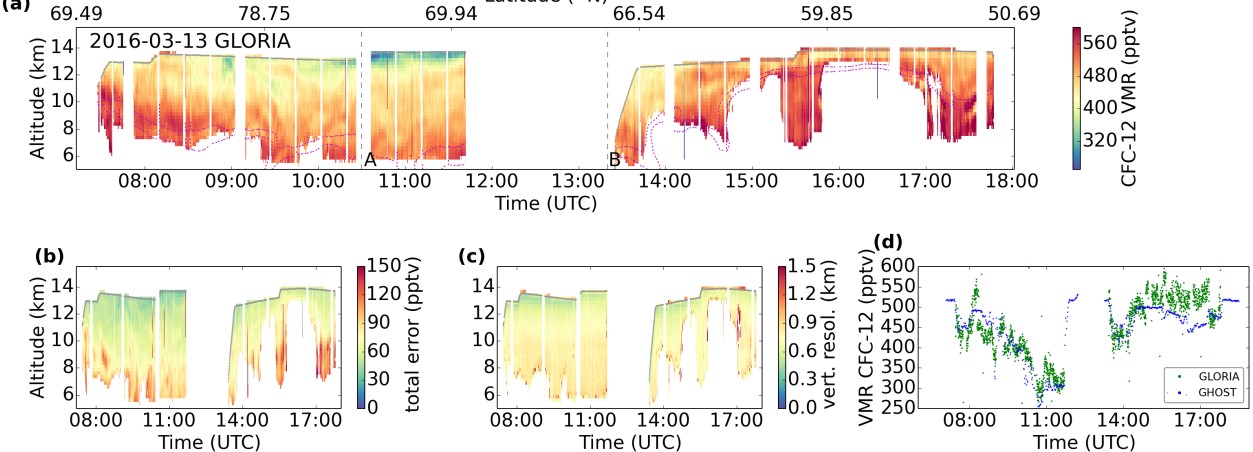

**Figure 11.** CFC-12 flight PGS19: Cross section of (a) retrieved CFC-12 volume mixing ratio (the flight altitude is marked with a grey line, the ECMWF potential vorticities of 2 and 4 PVU are marked with magenta dashed lines, way points are marked with grey vertical dashed lines). Cross sections of (b) total estimated error, (c) vertical resolution and (d) comparison of the GLORIA measurements (green) to the GhOST-MS in-situ measurement (blue).

## 4.4 Overview of in-situ comparisons for the PGS campaign

For an overview of comparisons of GLORIA high spectral resolution retrieval results to in-situ measurements for all PGS flights, the median difference and the median absolute deviation (Rousseeuw and Croux, 1993) are presented in Tab. 2. The median gives a measure of the accuracy of the match between the data sets, and the median absolute deviation is a method for describing the spread around this median value. Both measures are robust methods and for that reason a few extreme mismatches do not have a large influence. Detailed plots for flights that are not described as thoroughly as the flight PGS19 are provided as a supplement. Those plots also help to understand larger deviations (e.g. in temperature) between GLORIA and in-situ measurements that are present in numerous flights in January 2016, which have been strongly affected by PSCs at and above flight level.

The comparisons of GLORIA and in-situ instruments over the whole campaign show that there are reasonably low biases between the data sets. Atmospheric conditions that influence the measurement conditions for remote sensing change during the winter: In January many PSCs occur, which influence the measured infra red spectra and make temperature and trace gas retrievals challenging. Towards the end of the Arctic winter, more delicate structures in trace gases are present due to nitrification and related events, which make comparisons of measurements at different geolocations more difficult. These changing atmospheric conditions are also visible in the comparisons in Tab. 2: Deviations between GLORIA and BAHAMAS temperatures are larger for flights in January (due to the influence of PSCs). MLS $O_3$ and $HNO_3$ values become increasingly smaller compared to the corresponding GLORIA measurements towards the end of the Arctic winter. This is explained by the



**Table 2.** Median differences between GLORIA and in-situ and Aura/MLS measurements with the median absolute deviation (as a measure of the spread of the difference around the median value) for each flight and the whole campaign. For the flight on 12 January 2016 (PGS06) no water vapor in-situ measurements are available.

| Flight date | Temp. [K] BAHAMAS | $HNO_3$ [ppbv] AIMS | $HNO_3$ [ppbv] MLS | $O_3$ [ppbv] FAIRO | $O_3$ [ppbv] MLS | $ClONO_2$ [pptv] AIMS | $H_2O$ [ppmv] FISH | CFC-12 [pptv] GhOST |
|---|---|---|---|---|---|---|---|---|
| 15-12-21 | -0.97 ± 0.63 | 0.38 ± 0.33 | 0.99 ± 0.35 | 40.5 ± 89.9 | 197.4 ± 86.9 | 20.3 ± 64.9 | -0.42 ± 0.52 | -21.9 ± 25.6 |
| 16-01-12 | -1.15 ± 0.91 | 0.03 ± 0.68 | 1.31 ± 0.79 | -123.4 ± 127.4 | 257.4 ± 176.7 | -35.3 ± 84.5 | | -58.2 ± 21.7 |
| 16-01-18 | -2.04 ± 1.47 | 0.47 ± 1.15 | 1.39 ± 0.82 | 66.6 ± 80.2 | 261.0 ± 149.1 | -11.2 ± 75.8 | -0.67 ± 0.59 | -66.5 ± 50.5 |
| 16-01-20 | -1.99 ± 1.21 | -1.03 ± 0.80 | 1.41 ± 0.74 | 19.7 ± 129.0 | 266.4 ± 126.6 | -33.3 ± 70.7 | -0.71 ± 0.76 | -44.6 ± 48.3 |
| 16-01-22 | -1.09 ± 1.15 | -0.10 ± 1.30 | 1.83 ± 0.66 | -21.2 ± 108.9 | 365.7 ± 101.2 | -11.0 ± 84.3 | -0.01 ± 0.70 | -55.3 ± 46.1 |
| 16-01-25 | -2.18 ± 0.66 | -0.82 ± 0.99 | 1.84 ± 0.63 | 4.5 ± 109.9 | 435.8 ± 170.3 | -6.6 ± 85.3 | -0.32 ± 0.49 | -57.9 ± 29.4 |
| 16-01-28 | -1.78 ± 0.67 | -0.48 ± 0.42 | 1.37 ± 0.58 | 4.4 ± 37.8 | 230.4 ± 104.3 | 56.0 ± 78.6 | -0.69 ± 0.40 | -28.0 ± 13.7 |
| 16-01-31 | -0.56 ± 0.60 | -1.83 ± 1.52 | 2.26 ± 0.86 | 17.8 ± 76.9 | 397.1 ± 134.6 | 7.3 ± 67.6 | -0.75 ± 0.52 | -9.2 ± 26.2 |
| 16-02-02 | -0.45 ± 0.76 | 0.21 ± 0.79 | 1.94 ± 0.72 | 17.0 ± 50.7 | 324.0 ± 123.7 | -17.4 ± 60.2 | 0.07 ± 0.64 | -17.1 ± 32.3 |
| 16-02-26 | -0.98 ± 0.80 | 0.23 ± 0.66 | 2.12 ± 1.58 | -42.4 ± 92.4 | 319.8 ± 210.3 | -22.4 ± 88.9 | -0.24 ± 0.68 | 7.6 ± 29.2 |
| 16-03-06 | -0.82 ± 0.66 | 0.10 ± 0.48 | 2.46 ± 1.26 | 12.8 ± 68.6 | 387.2 ± 194.5 | 1.7 ± 87.4 | -0.16 ± 0.68 | -21.9 ± 53.6 |
| 16-03-09 | -0.83 ± 0.74 | -0.38 ± 0.88 | 2.64 ± 1.06 | 72.9 ± 144.2 | 418.3 ± 194.0 | -35.0 ± 127.8 | -0.05 ± 0.49 | -41.8 ± 49.9 |
| 16-03-13 | -0.14 ± 0.86 | -0.24 ± 0.95 | 3.04 ± 1.10 | 55.7 ± 139.6 | 465.3 ± 225.2 | -18.4 ± 202.6 | 0.24 ± 0.55 | 2.4 ± 39.6 |
| 16-03-16 | -0.06 ± 0.78 | 0.02 ± 0.42 | 0.32 ± 0.53 | -114.8 ± 120.1 | 113.2 ± 97.8 | -9.9 ± 235.4 | -0.06 ± 0.55 | 26.4 ± 41.0 |
| 16-03-18 | -0.64 ± 0.98 | 0.69 ± 0.60 | 3.57 ± 0.76 | 60.5 ± 152.8 | 549.0 ± 234.4 | -47.0 ± 131.5 | 0.19 ± 0.53 | 18.1 ± 34.6 |
| Campaign | -0.75 ± 0.88 | -0.03 ± 0.85 | 2.01 ± 1.33 | -3.5 ± 116.8 | 346.0 ± 202.7 | -15.4 ± 102.8 | -0.13 ± 0.63 | -19.8 ± 46.9 |

fine structures which are visible in the GLORIA measurements, but not resolved in Aura/MLS data due to their lower vertical resolution and horizontal gridding.

## 5 Conclusions

We discuss recent a survey of recent measurements in the high spectral resolution mode of the imaging FTS limb sounder GLORIA, which was deployed on the German research aircraft HALO during the PGS field campaign in the Arctic winter 2015/2016. As an example, we discuss the flight PGS19 on 13 March 2016 in detail, showing the retrieval results of temperature and the trace gases $HNO_3$, $O_3$, $ClONO_2$, $H_2O$ and CFC-12 and compare them to in-situ measurements and to MLS where applicable. We demonstrate that valuable information at high spatial resolution can be retrieved from infrared limb imaging data even in the UTLS with high clouds and PSCs present. Fine vertical structures can be examined thanks to vertical resolutions of 400 to 1000 m. Typical estimated errors are in the range of 1 - 2 K for temperature, and 10 - 20 % relative error for the discussed trace gases. An approach for post-flight LOS correction was successfully established to account for limited in-flight



10    LOS knowledge and stabilization due to technical and software problems.

The comparisons of the MLS and GLORIA HNO$_3$ and O$_3$ measurements show the advantage of airborne measurements: The aircraft measurements with high spatial resolution reveal small-scale structures in the trace gas distributions. In contrast, the satellite measurements provide a continuous time series of global measurements up to high altitudes, which helps to put the structures observed by GLORIA into context. The qualitative comparison shows that the same structures in O$_3$ and HNO$_3$ are visible in both data sets and the measured mixing ratios are of the same order of magnitude. Towards the end of the winter, O$_3$ and HNO$_3$ are underestimated by MLS, which is an effect of lower vertical resolution of the spaceborne instrument and horizontal gridding. This lower resolution does not resolve spatially confined enhancements in these trace gases.

Comparisons of the GLORIA retrieval results with in-situ measurements on board HALO show the consistency of these data sets, taking into account the error, vertical and horizontal resolutions of GLORIA and atmospheric variability which are pronounced by the different measurement techniques and the inferred different geolocations of the measurements.

This newly presented GLORIA data set benefits from aero-acoustic improvements of the instrument compared to a previous GLORIA campaigns (Kaufmann et al., 2015; Woiwode et al., 2015; Ungermann et al., 2015). It is based on a much higher

number of measured profiles and also has been compared to additional in-situ trace gas measurements. Compared to the data set by Woiwode et al. (2015), which is also based on measurements in the high spectral resolution mode, the vertical resolutions of this GLORIA data set are significantly better and a more detailed approach for error estimation is introduced. Furthermore, GLORIA measurements discussed in the paper at hand provide temperature and trace gas information down to 5 km, which is lower compared to the majority of previously discussed infrared limb sounders. In future retrieval setups we will aim to retrieve

additional trace gases such as C$_2$H$_6$ and PAN.

The results demonstrate the performance and quality of this GLORIA data set of the UTLS during the Arctic winter 2015/2016. GLORIA measurements of this region with rather high spatial sampling allow further studies on chlorine deactivation, denitrification and mesoscale structures.

*Data availability.* The discussed GLORIA data set and in-situ data sets are available at the HALO database https://halo-db.pa.op.dlr.de. Aura/MLS data is available at https://mls.jpl.nasa.gov. ECMWF analysis data is available at https://www.ecmwf.int.

*Competing interests.* The authors declare that they have no conflict of interest.

5     *Acknowledgements.* We gratefully thank the POLSTRACC/GW-LCYCLE II/SALSA coordination team and DLR-FX for successfully conducting the field campaign. The results are based on the efforts of all members of the GLORIA team, including the technology institutes ZEA-1 and ZEA-2 at Forschungszentrum Jülich and the Institute for Data Processing and Electronics at the Karlsruhe Institute of Technology. We thank the European Centre for Medium-Range Weather Forecasts (ECMWF) for providing their meteorological analyses. S. Johansson has received funding from the European Community's Seventh Framework Programme (FP7/2007-2013) under grant agreement





10  603557. A. Marsing, T. Jurkat-Witschas and C. Voigt were funded by DFG under contract number JU 3059/1-1 and VO 1504/4-1. This work
    was partly supported by the Bundesministerium für Bildung und Forschung (BMBF) under the project ROMIC/GW-LCYCLE, subproject
    (01LG1206B). S. Johansson gratefully thanks the Graduate School for Climate and Environment (GRACE), Karlsruhe Institute of Tech-
    nology for funding his visit to the Jet Propulsion Laboratory to discuss the comparisons with the Aura/MLS measurements and the MLS
    team for the hospitality during that time. Work at the Jet Propulsion Laboratory, California Institute of Technology, was done under contract
with the National Aeronautics and Space Administration. We acknowledge support by the Deutsche Forschungsgemeinschaft and the Open
    Access Publishing Fund of the Karlsruhe Institute of Technology.



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
