# Peer review of "Airborne limb-imaging measurements of temperature, $HNO_3$ , $O_3$ , $ClONO_2$ , $H_2O$ and CFC-12 during the Arctic winter 2015/16: Characterization, in-situ validation and comparison to Aura/MLS"

_Atmospheric Measurement Techniques, 2018_

## Referee Comment (RC1) · Anonymous Referee #1 · 28 May 2018

The manuscript reports the results of high spectral resolution atmospheric emission measurements of UTLS temperature and composition (HNO3, O3, ClONO2, H2O and CFC-12) conducted by the limb-imaging Fourier transform infrared spectrometer GLO-RIA on board the HALO research aircraft during the Arctic winter 2915/16. A clear overview is presented of the substantial amount of data acquired by GLORIA in the 15 flights of the PGS campaign, with a focus on the analysis of a single flight supported by detailed insight into the retrieval process and systematic and noise error

estimate. Validation of GLORIA temperature and trace gases retrieved profiles is performed against correlative data by the HALO in-situ payload and comparison with the Aura/MLS space-borne observations. Along with an appendix showing plots of results from the other flights, the paper includes an extensive list of bibliographic references and information for access to the discussed data sets from GLORIA, from the HALO in-situ payload, from Aura/MLS and ECMWF analysis, in line with the declared purpose to serve as a baseline for scientific studies using GLORIA measurements.

GENERAL COMMENTS The paper is well written and properly organized and its contents fit the scope of the AMTD/AMT journal. The material presented by the authors is sufficiently informative and significant new results are pointed out in specific sections of the paper and consistently highlighted in the conclusions. The relevance of improvements in the quality of the retrieval products and in the extended coverage of atmospheric targets resulting in the retrieval products from instrument upgrading with respect to measurement performance described in previous publications cited in the references is unquestionable. The need for accurate high vertical resolution measurements of critical regions of the Earth's atmosphere such as the Upper Troposphere and Lower Stratosphere is made explicit in the article and that need, which can be achieved only by limb-sounding instruments, has become even stronger due the warning for the risk of the limb-gap in spaceborne observations. I recommend the paper for publication after technical corrections and minor changes.

TECHNICAL CORRECTIONS The statement "Space-borne measurements provide global coverage" (pag. 2, line 9) is not necessarily true. Geostationary satellites do not provide global coverage. I suggest the following modification: "Space-borne measurements can provide global coverage"

The detailed description of the results from a single flight from the PGS campaign was performed by selecting flight PGS-19. Is that the result of a purely random choice or of a selection based on pre-established criteria? A short statement providing this information to the reader might be of help.

[Figure]

The term "combination" (suggesting a synergistic use of data) referred to the link established between GLORIA and MLS data does not appear the most appropriate. The extent to which the two datasets were jointly used to build the results reported in the manuscript appears to be rather limited. The term "comparison" might still be more appropriate to represent the actual exercise conducted using both data sets. I leave to the authors to decide on this point.

If available from the diagnostics of GLORIA measurements during the PGS campaign (or the PGS-19 flight): which is the typical amount of bad pixels filtered out (per row or per image)? Is that affecting the quality of the measurements in a significant manner with margins for future improvenets?

In the statement "Another important quantity for a retrieval is the degrees of freedom" (pag.9, line 26), the correct expression to use is "the number of degrees of freedom".

The statement "... since the diagonal element of each averaging kernel row isa measure ... retrieval results" (pag. 9, line 27) shall be formulated in a different manner, to avoid using the expression "diagonalelement of a row".

MINOR CHANGES I recommend the authors to revise the use of commas throughout the manuscript. An extensive (but not exhaustive) list of this kind of modifications is included among the minor changes here below.

Pag 1, line 15 – insert comma after "Additionally" Pag 2, line 29 – change "gases" to "gas" Pag. 3, line 3 – change "de-/activation" to "de-activation" (here and elsewhere in the paper for both de-/activation and de-/nitrification) Pag. 3, line 6 – insert comma after "purpose" Pag. 3, line 24 – insert comma after "In this paper" and after "In this measurement". Pag. 5, line1 – insert comma after "system" Pag. 5, line27 – change "auto-co-variance Function analysis" to "auto-covariance function analysis" Pag. 6, line23 – insert comma after "model" Pag. 6, line 30 – use the same spelling for a priori here ("a priori") and at pag. 7, line 3 ("a-priori") Pag. 7, line14 – insert comma after "purpose" Pag. 7, line17 – insert comma after "work" Pag. 7, line21 – insert comma

after "step" Pag. 7, line22 – insert comma after "retrieval" Pag. 8, line3 – insert comma after "H20" Pag. 9, line 4 – change ":" to "." Pag. 9, line6 – insert comma after "method" Pag. 9, line 14 – insert comma after "Here" Pag. 11, line 5 – change "For this reason no comparisons . . . are shown" to "For this reason, no comparison . . . is shown". Pag. 11, line 9 - insert comma after "track" Pag. 11, line 24 - insert comma after "Fig. 4" Pag. 11, line 26 - insert comma after "HNO3" Pag. 14, line 28 - insert comma after "reason" Pag. 15, line 6 - insert comma after "temperature" Pag. 15, line 11 - Include the statement "which is closest . . . better than 2 km" between two commas. Pag. 16, line 1 - change "flight PGS19 are presented in Fig. 7" to ""Flight PGS19 is presented in Fig. 7" Pag. 18, line 7 – insert comma after "(Clarmann et al., 1993; Oelhaf et al., 1994; Roche et al., 1994)" Pag. 20, line 10 – insert comma after "(way points "A" to "B")". Pag. 22, line 4 – remove "recent" after "we discuss" Pag. 23, line 10 – insert comma after "atmospheric variability".

---

## Referee Comment (RC2) · Anonymous Referee #2 · 14 Jun 2018

GENERAL COMMENTS:

The paper reports on the analysis of a large number of airborne measurements of atmospheric composition, taken by the infra-red limb-imaging Fourier Transform Spectrometer GLORIA during a number of flights campaigns in winter 2016/2017. The paper aims to be the reference document for all the 15 research flights of the three sub-campaigns in the overarching PSG campaign activity. It does this by providing a

detailed analysis of all geophysical observables for flight number PSG 19 on 2017-03-13, and an overview of key findings from the remaining flights. The reason for selection of flight 19 is not explicitly given, but we guess this is likely because the combination of instrument performance and the prevalence of interesting atmospheric features was best for this day. Conversely, another flight (PSG 12 or 2016-03-31) is occasionally referenced, but not in as much details as PSG 19. For the other flights only a table with the median deviations and variances of the retrieved parameters with respect to the in-situ validation is given. This is mostly adequate, although some references to specific features in other flights are found in the text (i.e. on the influence of PSCs), which are therefore not illustrated (See Technical Comments).

The paper offers a very comprehensive literature reference to previous measurements campaigns by the GLORIA instrument, and it covers the changes/improvements both in instrument performance, as well as the performance of the retrieval processor. The latter includes a description of an improved handling of pointing error in the Level1 processor, which is a major component of the overall error budget. The paper is clearly structured in relevant sections, describing the campaign parameters and flight performance, the instrument performance (including references to the correlative in-situ validation datasets), the data processing scheme (retrieval processor), and a detailed overview of the measurements, errors, and comparisons to validating datasets from both satellite measurements (Aura MLS) and in-situ instruments on the same flight platform. This is followed by a short summery of the measurement performance from the remaining flights.

In summary: The paper is structured in a logical fashion. It covers all the topics one expects to be discussed in a reference paper on a multi-flight campaign with an updated instrument. It is generally well written, the Figures are of a high quality, and a comprehensive list of references is included, which allows the work to be put in the right context. My opinion is that the paper should be published in AMT, if the issues related to content and substance listed in the paragraph "Technical Comments" are resolved.

It would also be desirable if the minor corrections on format and phrasing would be address, as this would make the paper more easily legible. Corrections are tagged by [page number/line number].

TECHNICAL CORRECTIONS:

[1/13] ". . . differences are mainly within the expected performance" "Event with stronger deviations are explained. . .". You need to quantify where you set the threshold between what you consider an "acceptable" overlap, and the onset of "unfavourable conditions" which consequentially prohibit a direct comparison. (On a sidenote, "mostly" would be better than "mainly" as you're describing a countable factor, but in general phrases like "mostly, mainly, or more or less" should be avoided in a scientific paper if at all possible.

[2/10] "Space-borne measurements. . .are limited in sampling and accuracy". Maybe say: "Current space-borne measurements. . ." to acknowledge the next generation of instrument, i.e. AtmoSat that will do much better.

[2/28] ". . .showed reasonable agreement. . .". Again, be specific. What does 'reasonable' mean, and how does the 'stage of development' affect this?

[3/2] "The scientific objectives. . .". This sentence/list is too long. It gets confusing. Why not write: "Among the scientific objectives of PSG campaign are: . . . ; . . . ; . . . ; and . . . Importantly, there should be a comma after "chlorine de-/activation[,] and de-/nitrification" or else the sentence implies that there is "chlorine de-/nitrification".

[3/14] ". . .corresponds to a displacement of the carrier. . .". Don't call the aircraft a carrier, call it the "aircraft", or the "platform". The expression 'displacement of the carrier' could be confused with the movement of carrier for the roof mirror inside your FTS instrument.

[4/25] ". . .onto the correct abscissa in space.". I don't understand this. Is this to correct for spherical aberration in a Gauss beam?

[4/27] ". . .different temperatures.". What temperatures? Are you using cooled, heated

or ambient targets? The temperature differences between the calibration targets, and their relation to the Brightness Temperature in the limb will affect your calibration errors (mainly the gain).

[4/Fig.1] The colours in the legend are unclear. I.e. I can't tell PSG19-21 apart, which is critical because PSG 19 is your main flight. Also, on the legend there are at least 4 flights in different hues fo blue, but I can only see 1 blue track on the map. Incidentally, you also refer to flight PSG 12 on several occasions in the paper (i.e. Fig.2) so you should probably highlight this one as well on the map.

[5/17] "...a precision of 0.7% x VMR +/- 0.35ppmv". I'm not sure this makes sense. In a format X +/- Y, Y is the "precision", so how can you have a value for precision that has itself a precision attached to it? I guess you're talking about a statistical analysis of an ensemble of measurement precisions. If so maybe worth to clarify.

[5/30] "precision of X and an uncertainly of Y". Again, same as above: "uncertainly" = "precision"

[6/22] "...radiation transfer model ... optimized for highly resolves spectra". I think they are generally called "radiative" transfer models. Also, the spectral resolution of a RTM is usually constrained by computational resources alone, not the algorithm, so I don't understand how the RTM can be "optimized" for high resolution.

[7/9] "... a constant (H2O) profile of 10ppmv is used". Even in the Troposphere? That could have a big impact on your simulated radiances because it would significantly change the opacity at the far end of your pencil beam (Tropospheric 'continuum').

[7/16] "calibration errors" and "pointing errors" are listed twice.

[9/7] "With this method...". I find this entire sentence confusing. Radiometric calibration errors are not attributed to gain and offset, but they result in gain and offset errors. They are attributed to things like errors in temperature knowledge, non-blackbody emissivities, standing waves, etc.

[9/9] "LOS errors are estimated...". Again, I don't fully understand what you did here. This is important, as the handling of LOS errors are a dominant error source according to you, so it needs to be crystal clear how they have been handled. (i.e. is the 0.05deg perturbation the variance of all unperturbed profile retrievals?)

[9/13] "...the related temperature error". I presume this is the T error in the ECMW data?

[9/28] "... the diagonal element of each averaging kernel row...". This is an incorrect definition of the degrees of freedom in the retrieval. To start with, a vector (AVK row) can't have a diagonal element per definition. Please review!

[10/16] "This stop allowed for higher altitudes of the HALO aircraft...". It's not the stop that makes the plane fly higher. How about: "HALO reaches its peak ceiling altitude immediately before each refuelling stop, when the airframe is at its lightest. It's only at these phases of the flight that the flight altitude is high enough to sample subsided polar...".

[12/Fig.3] The flight track in the vicinity of waypoints A and B is not very visible. Could you use lighter colours?

[13/Fig.4] The axes of panels c) and f) (vertical resolution) should be capped at 1.5km (or even 1km instead of 3.0km. This would better resolve the profile variations at the altitudes that actually matter.

[14/2] "...caused by changes in the atmospheric state...". Why is that? Changes in refracted path if the temperature/density is incorrect?

[14/8] "For flights between..." and following sentences: I think I understand what you are saying, but I had to read this section many times over before it became clear to me. Could you rephrase it in a less convoluted way?

[15/Fig. 5]: I can't tell the dark blue and the black dots apart in my A4 printout. Please use high contrast colours, i.e. red and black.

[17/4] "This is the same regions, where the HNO3...". I have the impression that the HNO3 mismatch peaks at 17:00h, while the O3 mismatch peaks at 16:00h. Is that really the same air-parcel? On the same note: Why are AIMS and FAIRO comparisons plotted on what are really quite different time-scales in their respective panels.

[18/?] "Baffin Bay". Where is Baffin Bay located in Fig 3? Not really common knowledge.

[18/?] General comment to this paragraph: You really should mention the good agreement between spatial features observed in O3 and HNO3. This is what you would expect from atmospheric chemistry, and the fact that you actually see it is an important self-validation of your results!

[18/7] "...subsided deactivated ClONO2". A large presence of the reservoir gas ClONO2 is a sign of "deactivated" ClO, and should therefore probably be called "activated" ClONO2.

[21/22] "... numerous flights in January 2016, which have been affected by PSCs...". This merit a separate Figure, and a short paragraph. It constitutes a separate, unique scientific finding of the campaign. Because the paper is aiming to be the reference publication of all flight in PSG, this should not be demoted to a mere side-not, just because it's not visible in the example flight PGS 19. You make a reference to a "supplement" that contains these additional plots, which I presume are part of the special edition, but I don't have access to this supplement, and neither will anyone that downloads your article as a standalone document form a research database a few years down the line. To make the paper useful in the long term, this link should either be omitted, or at the very least you will have to reference it (with full DOI information) in the text.

[23/17] "This lower resolution does not resolve spatially confined enhancements in these trace gases." MLS is still sensitive to the filament enhancements, even if it can't resolve them. If you re-grid the data carefully, i.e. by applying the MLS averaging kernels to your measurements, the observed VMR values should match. Are you comparing the peak VMR values from a high vertical resolution IR measurements with a low vertical resolution MSR measurement? In this case, the discrepancy is indeed to be expected, but it's not strictly because the MLS can't see it, but because the MLS measurements contains information from (O3-depleted) polar stratospheric air. So, you're comparing apples with pears.

[23/13] "...which is lower compared to... previous IR limb sounders". Why is that?

[23/17] Again, "rather" is a very meek and unspecific term. Your closing sentence should have some clout. How about: "GLORIA measurements with unprecedented spatial resolution over the Arctic region will form the basis for many future case studies on ..."

FORMAL (MINOR) CORRECTIONS:

General remarks: - Use of punctuation is not always consistent, especially commas - Use a digit divider for large numbers (i.e. 15'000, or 15,000. Pick whatever you prefer, but be consistent) - Be consistent with ligatures (i.e. limb-scanning vs. limb scanning,

[Title] Capitalisation of "Characterisation" after a colon.

[1/14] "stronger" "larger"

[2/1] Repetition of "region of". Capitalisation of Upper Troposphere Lower Stratosphere.

[2/5] The acronym GLORIA has been defined before.

[2/22] Define STR in MIPAS-STR

[2/26] Define ESSenCe

[3/7] "These 18 PSG research flights, each with a duration of approximately 10 hours, cover the entire period of the Arctic winter and provide a unique dataset"

[3/14] Repetition of "48 x 128 interferograms are recorded every 13 s"

[5/3] "enhanced" "'improved"

[5/11] "also allow" "assure"

[5/13] ". . . a detection limits of X, a precision of Y, and an accuracy of Z"

[5/16] Missing "is" in "which [is] based on. . ."

[6/6] Convoluted description of the MLS viewing geometry. Just say: "The along track scanning radiometers scan the limb every 165km."

[8/18] You explicitly reproduce the numbers of all errors from the literature, except for O3 and H2O line intensities (Flaud et al.). Would be more consistent to include them as well.

[10/27] HALO "left" or "exited" these air masses ("departed" implies an untimely demise of the aeroplane). "decreasing down to" "decreasing to".

[10/31] ". . .for dates and times of PSG flights, filtered by. . ."

[11/37] "to significantly change" "to vary significantly"

[11/11] "It can be seen that the shapes of the . . . profiles differ significantly from. . ."

[11/19] "most" "mostly", or better "predominantly"

[11/27] "This simultaneously illustrates the amount of data that. . . , and allows the characteristics. . ."

[15/12] Confusing, because high/low applies to both T and altitude in the same sentence. Maybe: ". . .from higher temperatures (240K) at lower altitudes, down to temperatures as low as 205K at flight altitude".

[16/17] Remove "likewise"; It has no meaning in this sentence.

[16/21] "implied" "visible", "evident", "shown"

[19/8] Typo: "stratopsheric" "stratospheric"

[21/18] Missing word: "median [difference] gives"

---

## Author Comment (AC1) · 10 Jul 2018

We thank referee 1 for valuable comments and suggestions. Our answers are given below. The original referee comment is repeated in **bold**, changes in the manuscript text are printed in *italic*.

**TECHNICAL CORRECTIONS:**

[Figure]

**The statement "Space-borne measurements provide global coverage" (pag. 2, line 9) is not necessarily true. Geostationary satellites do not provide global coverage. I suggest the following modification: "Space-borne measurements can provide global coverage"**
We changed the manuscript according to the referee's suggestion.

**The detailed description of the results from a single flight from the PGS campaign was performed by selecting flight PGS-19. Is that the result of a purely random choice or of a selection based on pre-established criteria? A short statement providing this information to the reader might be of help.**
We added the following sentence in Sec. 4, to briefly motivate the selection of flight PGS19 for detailed analysis: *Flight PGS19 is selected as an example of continuous measurements in high spectral resolution mode, and as an example of an illustrative amount of atmospheric variability within the measured air masses.*

**The term "combination" (suggesting a synergistic use of data) referred to the link established between GLORIA and MLS data does not appear the most appropriate. The extent to which the two datasets were jointly used to build the results reported in the manuscript appears to be rather limited. The term "comparison" might still be more appropriate to represent the actual exercise conducted using both data sets. I leave to the authors to decide on this point.**
We changed the term to *comparison* according to the referee's suggestion.

**If available from the diagnostics of GLORIA measurements during the PGS campaign (or the PGS-19 flight): which is the typical amount of bad pixels filtered out (per row or per image)? Is that affecting the quality of the measurements in a significant manner with margins for future improvements?**
The amount of bad pixels is in the order of 5 to 10%. The bad pixel filtering should

not significantly degrade the retrieval result, as the noise error only plays a minor role (see Fig. 4). We change the text in Sec. 2.1 to: *For noise reduction, the pixels of each detector row are averaged after filtering of bad pixels (typically 5 to 10%).*

**In the statement "Another important quantity for a retrieval is the degrees of freedom" (pag.9, line 26), the correct expression to use is "the number of degrees of freedom".**
We changed the manuscript according to the referee's suggestion.

**The statement "... since the diagonal element of each averaging kernel row is a measure ... retrieval results" (pag. 9, line 27) shall be formulated in a different manner, to avoid using the expression "diagonal element of a row".**
We change the formulation to: *... since the diagonal elements of the averaging kernel are measures of how much measurement information is contained in the retrieval result per level.*

We also thank referee 1 for the detailed language corrections, which helped us to further improve the manuscript.

---

## Author Comment (AC2) · 10 Jul 2018

We thank referee 2 for valuable comments and suggestions. Our answers are given below. The original referee comment is repeated in **bold**, changes in the manuscript text are printed in *italic*.

**TECHNICAL CORRECTIONS:**

[Figure]

**[1/13] "... differences are mainly within the expected performance" "Event with stronger deviations are explained ... ". You need to quantify where you set the threshold between what you consider an "acceptable" overlap, and the onset of "unfavourable conditions" which consequentially prohibit a direct comparison. (On a sidenote, "mostly" would be better than "mainly" as you're describing a countable factor, but in general phrases like "mostly, mainly, or more or less" should be avoided in a scientific paper if at all possible.)**
We changed the text to: *73% of these differences are within twice the combined estimated errors of the cross-compared instruments. Events with larger deviations ..*
We also add a temperature/trace gas specific statement in Sec. 4.4: *Another measure for the agreement between GLORIA and in-situ instruments is the part of co-located measurements, of which the differences are within twice the combined estimated errors of the cross-compared instruments. For temperature 88%, for $HNO_3$ 73%, for $O_3$ 63%, for $ClONO_2$ 53%, for $H_2O$ 90%, for CFC-12 77%, and in total 73% of the comparisons show this agreement. $ClONO_2$, $O_3$, and $HNO_3$ show substantial variations at flight altitude (e.g. Figs. 9,8,7). We attribute the lower fraction of agreement to the higher atmospheric variability of those trace gases, thereby complicating the comparison due to the strongly differing instrumental sampling characteristics.*

**[2/10] "Space-borne measurements ... are limited in sampling and accuracy". Maybe say: "Current space-borne measurements ... " to acknowledge the next generation of instrument, i.e. AtmoSat that will do much better.**
We changed the manuscript according to the referee's suggestion.

**[2/28] " ... showed reasonable agreement ...". Again, be specific. What does 'reasonable' mean, and how does the 'stage of development' affect this?**
The Woiwode et al., 2015 paper is describing results from the first GLORIA field campaign. As for most newly constructed instruments, technical improvements were implemented after an analysis of the first results. The most impact on data quality had

modifications to reduce the aero-acoustical properties of aircraft and interferometer. We changed the text towards a more specific formulation: *..., showed an agreement with MIPAS-STR and in-situ instruments, within the profile-to-profile variations of GLORIA. After this campaign, aero-acoustical modifications of the aircraft and of the GLORIA instrument improved the precision of GLORIA measurements.*

**[3/2] "The scientific objectives ... ". This sentence/list is too long. It gets confusing. Why not write: "Among the scientific objectives of PSG campaign are: ... ; ... ; ... ; and ... Importantly, there should be a comma after "chlorine de-/activation[,] and de-/nitrification" or else the sentence implies that there is "chlorine de-/nitrification".**
We changed the manuscript according to the referee's suggestion.

**[3/14] "... corresponds to a displacement of the carrier ... ". Don't call the aircraft a carrier, call it the "aircraft", or the "platform". The expression 'displacement of the carrier' could be confused with the movement of carrier for the roof mirror inside your FTS instrument.**
We changed from *carrier* to *platform*.

**[4/25] "... onto the correct abscissa in space.". I don't understand this. Is this to correct for spherical aberration in a Gauss beam?**
Compared to the on-axis beam, the optical path difference (OPD) is shorter for radiation passing through the interferometer under an off-axis angle alpha: OPD(alpha)=OPD(0)*cos(alpha). Since the off-axis angle is different for each detector pixel on the array, the different OPD must be taken into account during the level 1 processing. We change the text to: *... and the optical path difference of each pixel is determined according to its off-axis angle, in order to sample each interferogram onto the correct abscissa in space.*

**[4/27] "... different temperatures.". What temperatures? Are you using cooled, heated or ambient targets? The temperature differences between the calibration targets, and their relation to the Brightness Temperature in the limb will affect your calibration errors (mainly the gain).**

The black-bodies can be either cooled or heated. In order to avoid ice contamiation, the cold black-body is kept only a few Kelvin below ambient temperature, while the hot black-body is heated to 30 to 40 K above the cold one. the higher radiance compared to the limb measurements is compensated by a lower integration time. We add a reference for the blackbodies and change the text to: *Gain and offset are determined from regular in-flight measurements of the two on-board black-bodies (Olschewski et al., 2013). The temperature difference between the two black-bodies is about 30 to 40 K with the cold black-body being around or slightly below ambient temperature.*

Olschewski, F., Ebersoldt, A., Friedl-Vallon, F., Gutschwager, B., Hollandt, J., Kleinert, A., Monte, C., Piesch, C., Preusse, P., Rolf, C., Steffens, P., and Koppmann, R.: The in-flight blackbody calibration system for the GLORIA interferometer on board an airborne research platform, Atmos. Meas. Tech., 6, 3067-3082, https://doi.org/10.5194/amt-6-3067-2013, 2013.

**[4/Fig.1] The colours in the legend are unclear. I.e. I can't tell PSG19-21 apart, which is critical because PSG 19 is your main flight. Also, on the legend there are at least 4 flights in different hues fo blue, but I can only see 1 blue track on the map. Incidentally, you also refer to flight PSG 12 on several occasions in the paper (i.e. Fig.2) so you should probably highlight this one as well on the map.**

It is difficult to find 17 colors, which are easy to distinguish for every-one, as colors are not perceived in the same way by different persons. Still, we tried a different approach with a color selection suggested by https://sashat.me/2017/01/11/list-of-20-simple-distinct-colors/ (checked 3 July 2018). We also highlighted flight PGS12 on the map as suggested.

**[5/17] "... a precision of 0.7% x VMR +/- 0.35ppmv". I'm not sure this makes sense. In a format X +/- Y, Y is the "precision", so how can you have a value for precision that has itself a precision attached to it? I guess you're talking about a statistical analysis of an ensemble of measurement precisions. If so maybe worth to clarify.**

The precision of the FISH instrument is estimated with a relative part (0.7% $\times$ VMR) and an absolute part (0.35 ppmv). We changed the manuscript to avoid the misleading +/- notation and to clarify: *FISH ... achieved a precision of 0.7% $\times$ vmr (volume mixing ratio; relative part of the precision) + 0.35 ppmv (absolute part of the precision) ... during PGS.*

**[5/30] "precision of X and an uncertainly of Y". Again, same as above: "uncertainly" = "precision"**

We replaced *uncertainty* by *accuracy (based on systematic errors)* in this sentence.

**[6/22] "... radiation transfer model ... optimized for highly resolves spectra". I think they are generally called "radiative" transfer models. Also, the spectral resolution of a RTM is usually constrained by computational resources alone, not the algorithm, so I don't understand how the RTM can be "optimized" for high resolution.**

We changed the text to: *..., which is optimized for computationally efficient analyses of highly resolved spectral measurements.*

**[7/9] "... a constant (H2O) profile of 10ppmv is used". Even in the Troposphere? That could have a big impact on your simulated radiances because it would significantly change the opacity at the far end of your pencil beam (Tropospheric 'continuum').**

We have tested this approach in comparison with the use of more realistic initial

guess profiles from ecmwf. The main effect was a larger number of iterations, but the retrieval results differed only within the estimated errors. Still we decided to use the constant initial guess and invest this larger number of iterations to be sure that any features in the retrieved profiles are not imposed by the initial guess. We added to the sentence: *... , a constant profile of 10 ppmv is used as initial guess, in order to assure independence of derived vertical and horizontal structures in the water vapor distribution e.g. compared to initial guess profiles from meteorological analysis.*

**[7/16] "calibration errors" and "pointing errors" are listed twice.**
Thank you for pointing that out. We removed the repetitions.

**[9/7] "With this method ... ". I find this entire sentence confusing. Radiometric calibration errors are not attributed to gain and offset, but they result in gain and offset errors. They are attributed to things like errors in temperature knowledge, non-blackbody emissivities, standing waves, etc.**
We changed this sentence to: *With this method, uncertainties in the radiometric calibration are calculated considering uncertainties in the multiplicative gain of 2% and uncertainties in the additive radiance offset of 50.0 nWcm$^{-2}$sr$^{-1}$cm.*

**[9/9] "LOS errors are estimated ... ". Again, I don't fully understand what you did here. This is important, as the handling of LOS errors are a dominant error source according to you, so it needs to be crystal clear how they have been handled. (i.e. is the 0.05deg perturbation the variance of all unperturbed profile retrievals?)**
The estimation of the 0.05° LOS perturbation is based on the short-term variance (not the long term changes!) of the LOS retrievals on a profile-to-profile basis (see Fig. 5b) and uncertainties within the retrieval itself. We changed the text to: *LOS errors are estimated by retrievals assuming a 0.05° LOS offset. This estimation is based on the short-term profile-to-profile variability found in the LOS retrievals (see Sec. 4.3.1),*

*and systematic uncertainties inherent to the LOS retrieval, such as uncertainties in ECMWF atmospheric temperature and pressure.*

**[9/13] "... the related temperature error". I presume this is the T error in the ECMW data?**
For trace gas retrievals, the retrieved temperature is used and thus also the temperature error related to this retrieved temperature. To clarify, we added: *... related temperature error (estimated for the temperature retrieval).*

**[9/28] "... the diagonal element of each averaging kernel row ... ". This is an incorrect definition of the degrees of freedom in the retrieval. To start with, a vector (AVK row) can't have a diagonal element per definition. Please review!**
We change the formulation to: *... since the diagonal elements of the averaging kernel are measures of how much measurement information is contained in the retrieval result per level.*

**[10/16] "This stop allowed for higher altitudes of the HALO aircraft...". It's not the stop that makes the plane fly higher. How about: "HALO reaches its peak ceiling altitude immediately before each refuelling stop, when the airframe is at its lightest. It's only at these phases of the flight that the flight altitude is high enough to sample subsided polar ... ".**
We changed the manuscript according to the referee's suggestion.

**[12/Fig.3] The flight track in the vicinity of waypoints A and B is not very visible. Could you use lighter colours?**
We changed the figure, such that the magenta flight track is put one layer above the white/transparent background of the way point labels to increase contrast at this region. At least in panel (a) the flight path is now easily visible, for panels (b)-(d) the

contrast still is somewhat lower due to the red colors of the shown meteorological/trace gas quantity. Still magenta is one of the few colors not included in the used colormap.

**[13/Fig.4] The axes of panels c) and f) (vertical resolution) should be capped at 1.5km (or even 1km instead of 3.0km. This would better resolve the profile variations at the altitudes that actually matter.**
We changed the figure according to the referee's suggestion.

**[14/2] "... caused by changes in the atmospheric state ... ". Why is that? Changes in refracted path if the temperature/density is incorrect?**
We clarify by stating the influence of temperature to the spectra of $CO_2$, which is also used for the LOS retrieval: *This difference in the retrieved LOS can be caused by differences in the atmospheric state compared to the ECMWF fields (which also affects the intensity of the $CO_2$ spectral lines, that are used for the LOS retrieval), ....*

**[14/8] "For flights between ... " and following sentences: I think I understand what you are saying, but I had to read this section many times over before it became clear to me. Could you rephrase it in a less convoluted way?**
We added a close-up of some of the discussed drifts and jumps in Fig. 5 to show the problem in a more detailed way. We extended the paragraph to: *For flights between 21 December 2015 and 31 January 2016 a software malfunction of the pointing control software caused the LOS to drift away from the commanded elevation. At certain points the software changed the instrument elevation back to its correct value and steep steps in the retrieved pointing elevation angle are observed in these flights (see Fig. 5a, enlargement: "Drift" and "Jump"). A correction of this artifact can be calculated by interpolating the LOS between the points immediately after a steep step. This interpolated line between the correct elevation angles approximates the LOS that would have been retrieved for a measurement without this software malfunction. The same average LOS correction, which is used for other flights, can be calculated from*

*this interpolated LOS (Fig. 5a, green points). This is the first part of the LOS correction for these flights. In the second part, the influence of the software malfunction can be extracted by subtraction of the interpolated LOS from the retrieved LOS. For an idealized measurement (without any further error in the LOS), this method separates the effect of the software malfunction from long-term variations (which have been corrected for in the first part). For subsequent retrievals of temperature and volume mixing ratios, both corrections, the average LOS correction and the correction of the steps, have been applied (Fig. 5a, red points).*

**[15/Fig. 5]: I can't tell the dark blue and the black dots apart in my A4 printout. Please use high contrast colours, i.e. red and black.**
We changed the figure according to the referee's suggestion.

**[17/4] "This is the same regions, where the HNO3 ... ". I have the impression that the HNO3 mismatch peaks at 17:00h, while the O3 mismatch peaks at 16:00h. Is that really the same air-parcel? On the same note: Why are AIMS and FAIRO comparisons plotted on what are really quite different time-scales in their respective panels.**
We adjusted all figures including in-situ comparisons to the same time-axis as for the GLORIA plots. The difference in the previous plots was caused by different data availability for each in-situ instrument, but we agree that for a better comparison the same time axis should be used.
With the newly adjusted time axis, it can be easier seen, that structures in $HNO_3$ and $O_3$ show similarities (for GLORIA and in-situ). For both trace gases, there is a decrease at 16:00 in GLORIA measurements, which is not seen in the in-situ data and also for both trace gases the structures shortly before 17:00 are reproduced by GLORIA. The amplitude of these structures show larger differences between remote sensing and in-situ for $HNO_3$ compared to $O_3$.

**[18/?] "Baffin Bay". Where is Baffin Bay located in Fig 3? Not really common knowledge.**

We clarified by adding to the text: *... Baffin Bay (the region covered by the GLORIA tangent points between way points "A" and "B"; ....*

**[18/?] General comment to this paragraph: You really should mention the good agreement between spatial features observed in O3 and HNO3. This is what you would expect from atmospheric chemistry, and the fact that you actually see it is an important self-validation of your results!**

We add the following part to the paragraph: *Spatial features are in agreement with the ones observed in $HNO_3$ (see Fig. 7), which is expected from atmospheric chemistry (Popp et al., 2009). This close correlation between the GLORIA measurements of both trace gases is an additional self-check for the validity of our results.*

Popp, P. J., et al. (2009), Stratospheric correlation between nitric acid and ozone, J. Geophys. Res., 114, D03305, doi: 10.1029/2008JD010875

**[18/7] "... subsided deactivated ClONO2". A large presence of the reservoir gas ClONO2 is a sign of "deactivated" ClO, and should therefore probably be called "activated" ClONO2.**

We changed the text to: *... subsided deactivated chlorine in form of $ClONO_2$.*

**[21/22] "... numerous flights in January 2016, which have been affected by PSCs ... ". This merit a separate Figure, and a short paragraph. It constitutes a separate, unique scientific finding of the campaign. Because the paper is aiming to be the reference publication of all flight in PSG, this should not be demoted to a mere side-not, just because it's not visible in the example flight PGS 19. You make a reference to a "supplement" that contains these additional plots, which I presume are part of the special edition, but I don't have access to this supplement, and neither will anyone that downloads your article as a**

**standalone document form a research database a few years down the line. To make the paper useful in the long term, this link should either be omitted, or at the very least you will have to reference it (with full DOI information) in the text.**
In Fig. 2a, flight PGS12 is shown as an example for a PSC affected flight. This was reported by the flight crew and is visible in the lower Cloud-Index values close to flight altitude. Still the CI does not proof the existence of PSCs, it only gives a measure for the cloudiness along the limb, which affects the trace gas retrievals. For a detailed PSC analysis, more advanced retrieval methods are necessary (e.g. Spang et al., 2016), which are out of the scope of the paper. Pitts et al. (2018) also give an overview of PSCs measured by CALIOP and the extension of PSCs down to lower altitudes in the 2015/16 Arctic winter are visible in his work.

To clarify, we change the text to: *...  numerous flights in January 2016, which have been strongly affected by PSCs at and above flight level. From the HALO flight crew, PSCs have been reported at these altitudes for PGS flights until PGS14 (26 February 2016). The influence of PSC and high altitude cirrus clouds on the spectra are shown in Fig. 2a as lower CI values at and below flight altitude.*

The supplement is publicly available via the AMTD website of this article: https://www.atmos-meas-tech-discuss.net/amt-2018-52/amt-2018-52-supplement.pdf.

According to Copernicus Publications, a DOI would be assigned to the supplement during the typesetting process in case of a publication in AMT.

Pitts, M. C., Poole, L. R., and Gonzalez, R.: Polar stratospheric cloud climatology based on CALIPSO spaceborne lidar measurements from 2006–2017, Atmos. Chem. Phys. Discuss., https://doi.org/10.5194/acp-2018-234, in review, 2018.

Spang, R., Hoffmann, L., Höpfner, M., Griessbach, S., Müller, R., Pitts, M. C., Orr, A. M. W., and Riese, M.: A multi-wavelength classification method for polar stratospheric cloud types using infrared limb spectra, Atmos. Meas. Tech., 9, 3619-3639, https://doi.org/10.5194/amt-9-3619-2016, 2016.

**[23/17] "This lower resolution does not resolve spatially confined enhance-**

**ments in these trace gases." MLS is still sensitive to the filament enhancements, even if it can't resolve them. If you re-grid the data carefully, i.e. by applying the MLS averaging kernels to your measurements, the observed VMR values should match. Are you comparing the peak VMR values from a high vertical resolution IR measurements with a low vertical resolution MSR measurement? In this case, the discrepancy is indeed to be expected, but it's not strictly because the MLS can't see it, but because the MLS measurements contains information from (O3-depleted) polar stratospheric air. So, you're comparing apples with pears.**
Unfortunately we have to stay with this more descriptive analysis since a quantitative comparison by applying the MLS AKs to the GLORIA measurements is not possible since GLORIA does not provide altitude resolved information above the flight level - where MLS AKs still have major contributions. In order to clarify, we change to: *This lower resolution does not resolve spatially confined enhancements in these trace gases. Due to only partial overlap of vertically resolved information from GLORIA and the width of the MLS averaging kernels, it is not possible to perform a more quantitative comparison.*

**[23/13] "... which is lower compared to ... previous IR limb sounders". Why is that?**
We add to the sentence: *... which is lower compared to the majority of previously discussed infrared limb sounders, due to the much higher vertical and horizontal sampling of the limb-imaging spectrometer.*

**[23/17] Again, "rather" is a very meek and unspecific term. Your closing sentence should have some clout. How about: "GLORIA measurements with unprecedented spatial resolution over the Arctic region will form the basis for many future case studies on ... "**
We changed the manuscript according to the referee's suggestion. Thanks for this excellent suggestion!

We also thank referee 2 for the detailed formal corrections. We applied most of these suggestions, which helped us to further improve the manuscript.
* * *